

# Gauge-Adjusted Rainfall Estimates from Commercial Microwave Links

Martin Fencl[1], Michal Dohnal[1], Jörg Rieckermann[2], Vojtěch Bareš[1]

[1]Department of Hydraulics and Hydrology, Czech Technical University in Prague, Prague 6, 166 29, Czech Republic
[2]Eawag: Swiss Federal Institute of Aquatic Science and Technology, 8600 Dübendorf, Switzerland

*Correspondence to*: Martin Fencl (martin.fencl@fsv.cvut.cz)

**Abstract.** Increasing urbanization makes it more and more important to have accurate stormwater runoff predictions, especially with potentially severe weather and climatic changes on the horizon. Such stormwater predictions in turn require reliable rainfall information. Especially for urban centers, the problem is that the spatial and temporal resolution of rainfall

observations should be substantially higher than commonly provided by weather services with their standard rainfall monitoring networks. Commercial microwave links (CMLs) are non-traditional sensors, which have been proposed about a decade ago as a promising solution. CMLs are line-of-sight radio connections widely used by operators of mobile telecommunication networks. They are typically very dense in urban areas and can provide path-integrated rainfall observations at sub-minute resolution. Unfortunately, quantitative precipitation estimates from CMLs (QPEs) are often

highly biased due to several epistemic uncertainties, which significantly limit their usability. In this manuscript we therefore suggest a novel method to reduce this bias by adjusting QPEs to existing rain gauges. The method has been specifically designed to produce reliable results even with comparably distant rain gauges or cumulative observations. This eliminates the need to install reference gauges and makes it possible to work with existing information. First, the method is tested on data from a dedicated experiment, where a CML has been specifically set up for rainfall monitoring experiments, as well as

many operational CMLs from an existing cellular network. Second, we assess the performance for several experimental layouts of "ground truth" from RGs with different spatial and temporal resolutions. The results suggest that CMLs adjusted by RGs with a temporal aggregation of up to one hour i) provide precise high-resolution QPEs (rel. error < 7 %, $R^2 > 0.75$) and ii) that the combination of both sensor types clearly outperforms each individual monitoring system. Unfortunately, adjusting CML observations to RGs with longer aggregation intervals of up to 24 h has drawbacks. Although it also

substantially reduce bias, it unfavourably smoothes out rainfall peaks of high intensities, which is undesirable for stormwater management. A similar, but less severe, effect occurs due to spatial averaging when CMLs are adjusted to remote RGs. Nevertheless, even here, adjusted CMLs perform better than RGs alone. Furthermore, we provide first evidence that the joint use of multiple CMLs together with RGs also reduces bias in their QPEs. In summary, we believe that our adjustment method has great potential to improve the space-time resolution of current urban rainfall monitoring networks. Nevertheless,

future work should aim to better understand the reason for the observed systematic error in QPEs from CMLs.



## 1 Introduction

Water-related issues are one of the major challenges of modern cities. Recently, more than 54 % of World's population lives in urban areas and the number is continuously growing (United Nations, 2014). Increasing urbanization, together with undergoing weather and climatic changes stresses the importance of efficient urban water management for preventing flooding and at the same time controlling pollution and ensuring sanitation. Rainfall is a main driver for many urban hydrological processes. Hence, reliable rainfall observations are crucial to informed decision making. Unfortunately, rainfall is very variable in both time and space, which makes it challenging to observe reliably. This is especially true for rainfall monitoring for urban stormwater management. Urban catchments usually consist of many small subcatchments with diverse land use characteristics. In cities, large fractions of impervious surfaces reduce the times of concentration and conduits, such as gutters, streets, etc., drain stormwater runoff very efficiently. Thus, runoff responses of urban catchments are usually very fast and greatly influenced by the spatial distribution and temporal dynamics of rainfall. Accurate predictions of rainfall-runoff therefore need rainfall information of high spatial and temporal resolution, which is difficult to get from point rain gauges (RGs) (Ochoa-Rodriguez et al., 2015).

Although the temporal resolution of RGs is adequate, the spatial representativeness of point rainfall observations is limited, especially for those heavy storm events which determine the design of urban stormwater systems. At many places around the world, S-band and C-band weather radars have therefore become an integral part of operational networks of weather and hydrological services. They can capture rainfall structure at the mesoscale, however, typical spatial and temporal resolution of radar's gridded precipitation product (usually 5 minutes and 1 km$^2$) is too low for urban hydrological applications (Ochoa-Rodriguez et al., 2015). In addition, radars measure rainfall hundreds of meters above ground (1 or 2 km of altitude at 100 km), due to the elevation of radar beam and Earth curvature (Berne and Krajewski, 2013). Finally, local weather radars, which are capable of providing rainfall observations at sub-kilometer/minute resolution, are rarely available. In addition, the data quality of quantitative precipitation estimates from radar in the heterogeneous urban environment can be compromised by many influences from the urban topology and morphology (Tilford et al., 2002). The extensive growth of GSM and other wireless networks in the recent decade around the globe opens new perspectives to improve urban rainfall monitoring with non-traditional sensors. These are either cheap simple sensors specifically designed for rainfall sensing (e.g. Stewart et al., 2012), or other devices which are disturbed by or detect rain and hence provide indirect rainfall observations, such as commercial microwave links (CMLs).

A CML is a point-to-point radio system which connects two remote locations. A CML features radio unit and a directional antenna transmitting a radio signal from one site (near-end) to another (far-end), where the signal is received by yet another unit. CMLs are commonly used by mobile network operators as a wireless connection in their backhaul network, but also by internet providers, military, and others. CMLs transmit electromagnetic waves, therefore rainfall intensities can be retrieved in a similar fashion as for weather radars. One important difference is, however, that a radar measures power of echoes



reflected by raindrops, whereas quantitative precipitation estimates from a CML (QPEs) are based on the rain-induced attenuation along its path (Atlas and Ulbrich, 1977).

Originally, the use of CMLs as rainfall sensors was suggested in the last century by Atlas and Ulbrich (1977). Interestingly, it has experienced a renaissance in the last decade with extensive growth of GSM network (Leijnse et al., 2007; Messer et al., 2006), and modern IT infrastructure, which makes it possible to actually collect data from hundreds or thousands of CMLs. First studies concentrated on algorithms for spatial temporal interpolation (Goldshtein et al., 2009; Overeem et al., 2013; Zinevich et al., 2008) from the joint analysis of multiple CMLs. Bianchi et al. (2013a, 2013b) have reported detection of malfunctioning RGs and improvement of radar observations by CMLs. The great potential of CMLs for ungauged regions was demonstrated by Doumounia et al. (2014). Interestingly, even though CML networks are most dense in urban areas, and thus are ideally suited for urban hydrological applications, there have been only very few investigations reported, which focus specifically on CML rainfall at the scale and resolution required for urban rainfall-runoff modeling (Fencl et al., 2013). A CML network in urban areas is usually very dense with many short hops (< 1 km) which have potential to capture rainfall with a high spatial resolution. On the other hand, network management systems are typically configured to monitor CML power levels once in 15 minutes, or even less often, which is insufficient for urban hydrological applications. Wang et al. (2012), however, showed that it is technically possible to poll CMLs with the sub-minute sampling frequency. Fencl et al. (2015) and Chwala et al. (2016) demonstrated the feasibility of this approach on a real network maintained by mobile operators. Unfortunately, the short CMLs are very sensitive to antenna wetting (Kharadly and Ross, 2001; Leijnse et al., 2008; Schleiss et al., 2013) which leads to substantial bias in their QPEs. Correcting this bias is, therefore, crucial for exploiting potential of CMLs for urban hydrology.

## 1.1 Biased rainfall estimates from commercial microwave links

Rainfall sensing with CMLs is based on relating the level of rain-induced attenuation to the rainfall intensity integrated along the CML path. As both rainfall intensity and attenuation are moments of the drop size distribution (DSD), the relation between attenuation and rainfall can be approximated by a power law:

$$R = \alpha k^{\beta}, \tag{1}$$

where $R$ [mm h$^{-1}$] is the rainfall intensity, $k$ [dB km$^{-1}$] is the specific path attenuation caused by raindrops and $\alpha$ [mm h$^{-1}$ km dB$^{-1}$] and $\beta$ [ - ] are empirical parameters depending on frequency, polarization of CML, and DSD (e.g. Olsen et al., 1978). For the frequency range of CMLs commonly used in cellular networks, the power law approximation leads to relatively low uncertainties in QPEs (Berne and Uijlenhoet, 2007), compared to the other uncertainties contributing to the specific path attenuation $k$ Eq. (1), which are associated with microwave propagation and CML hardware (Leijnse et al., 2008; Zinevich et al., 2010). Unfortunately, the microwave path propagation is not only influenced by raindrop scattering and adsorption, but also by a variety of other phenomena such as the refractivity of air, gaseous attenuation, etc. which are often not measured directly. In addition, the additional signal power loss caused by the wetting of the antenna surfaces, the so-called wet antenna attenuation (WAA) is causing a systematic overestimation of rainfall. Several WAA models have been





suggested to correct CML readings for this effect: from a simple empirically estimated offset (Overeem et al., 2011) to more complex semi-empirical models (Kharadly and Ross, 2001; Leijnse et al., 2008; Schleiss et al., 2013). Nevertheless, working with data from many hundreds of antennas, we experienced that the wetting and drying dynamics are complex processes which not only dependent on the individual antenna's material and characteristics (type and material of radome, surface

coating, orientation, exposure to wind, height over ground, etc.), but are also influenced by micro-weather and climate, such as local rainfall intensity, air humidity, wind speed and air temperature, to just name a few. Thus, it is generally difficult to correctly predict WAA for a specific CML because, i) our mechanistic understanding is limited and ii) important input data are not available. Last, but not least, the reliability of rainfall-induced path-attenuation is also compromised by inaccurate radio unit hardware, which measures transmitted ($Tx$) and received ($Rx$) signal levels of radio waves with a quantization of

up to 1 dB.

Such hardware-related influence factors are especially important for short CMLs. In general, CMLs shorter than 1 km could be potentially most informative for urban rainfall monitoring, because i) they could capture rainfall variability at the microscale and ii) their length corresponds with the dimensions of urban sub-catchments. Unfortunately, they are also less sensitive to rainfall, because they are comparably less attenuated by rainfall than long CMLs, simply because less scattering

occurs along the short path. Consequently, they are more sensitive to hardware-related errors (WAA and radio unit accuracy) which are path-length independent and thus contribute relatively more to the specific attenuation $k$ in Eq. (1) than the errors associated with microwave propagation. In the future, we might have detailed models to predict hardware related errors for each of the thousand CMLs of a commercial operator's network. Up until now, the most feasible approach in our view is to compare, and possibly adjust CML estimated rainfall with a ground rainfall observations to identify and eliminate systematic

errors in QPEs. However, to date there is no established method how to best achieve this goal.

As a first step, we reviewed the most relevant literature on adjusting rainfall radars. We found that i) most common adjustment methods are correcting the mean field bias of radar estimates to reference areal rainfall. The latter is usually calculated from point RG observations using a variety of interpolation methods (Smith and Krajewski, 1991), ii) the critical issue is the discrepancy between point RG observations, with a catch area of few dm$^2$, and areal rainfall estimated from radar

measurements with pixel sizes in the order of 1 km$^2$, iii) this discrepancy is typically reduced by using multiple RGs and also by rainfall aggregation over longer intervals, typically one hour (Wilson and Brandes, 1979).

In this paper, we employ these findings to suggest a method for continuous adjusting of commercial CMLs to cumulative rainfall from RGs. It is intended especially for urban catchments where, according to our experience, RGs are often available, but do not provide QPEs of sufficient resolution needed e.g. for reliable rainfall-runoff modeling. The main

novelty is that it is specifically tailored to the path-averaged attenuation of CMLs. Unlike radar reflectivity, this attenuation can be modelled by simplifying the power law of Eq. (1), which makes the adjusting procedure less prone to overfitting. Our results demonstrate that we can substantially reduce systematic errors from 50 % to about 7 %, which is very promising for the short CMLs in urban areas. In a fashion, our method can be viewed as a spatio-temporal disaggregation method for cumulative rain gauges based on the path-integrated high-frequent observations from CMLs. In our view, the combined use



of CMLs and RGs has, therefore, a very good potential to improve the space-time resolution of current local rainfall monitoring, which is of great importance for various applications in urban hydrology. Moreover, it can contribute to our deeper understanding of rainfall behavior at microscale and its implications for urban stormwater runoff.

The remainder of the paper is structured as follows: the Material and Methods section first describes the two experimental
sites, second, presents our suggestions to simplify the power law model and, third, how it can be conditioned to local RGs. We also discuss suitable statistics for performance assessment. Then, we present the results from two experimental sites, where in total five CMLs were adjusted by several different RG layouts. Finally, we discuss our approximation of the k-R relation together with issues of model calibration and overall limitations of the adjustment approach and draw our conclusions.

## 2 Material & Methods

This section first describes the experimental sites, their instrumentation, and the experimental period in terms of rainfall events. Second, a simplified attenuation-rainfall model is proposed together with a procedure how to continuously adjust its parameters. Finally, we suggest suitable statistics for performance evaluation.

### 2.1 Experimental sites

### 2.1.1 Dübendorf

The Dübendorf (CH) site represents an experiment where both CML and rainfall measurements were controlled to a high degree (Wang et al., 2012). The field campaign started in March 2011 and was maintained for more than one year. It consisted of a single commercial CML (MINI-LINK Ericsson) and an array of five laser precipitation disdrometers (Parsivel, OTT Hydromet, Germany) placed along the CML path (Fig. 1, right). The CML is a 38 GHz simple duplex dual polarized
link, i.e. the CML transmits and receives both vertically and horizontally polarized radio waves in both directions (from near end to far end and vice versa). It is 1850 m long originating at Dübendorf's military airport and ending at military radar site at Wangen. The CML path is located mainly over green surfaces of the airport and agricultural land. Here, we used data from a period where the automatic power control (ATPC), which maintains a constant received signal level ($Rx$) by adjusting the transmitted signal level ($Tx$) to minimize energy consumption and environmental radiation, was switched off. In addition to
the five disdrometers, three tipping bucket RGs measure rainfall intensities which makes it possible to validate the disdrometer data. For details, also on data retrieval via SNMP and pre-processing, see Wang et al. (2012) and Schleiss et al., (2013).



### 2.1.2 Prague-Letnany

In the Prague-Letnany (CZ) site, CMLs are an integral part of the existing cellular network and their operation is fully subordinated to its primary telecommunication function. The experimental catchment Prague-Letnany is a small urban catchment. The catchment area is 2.3 km$^2$, being approximately 2.5 km long in SN direction and 1 km wide in WE direction

(Fig. 1, middle). T-Mobile CZ, the mobile network operator which has kindly been supplying us with CML data, operates approx. 20 CMLs in the catchment. The CMLs are located approx. 40 m above ground level and their network mostly follows a star-shaped design. Current *Rx* and *Tx* levels are polled from each CML via the SNMP protocol using server-sided java script and stored in a SQL database (Fencl et al., 2015). CMLs are polled in serial sequence, each approximately 5 times per minute.

For the purposes of this study, we have selected four CMLs operating at frequencies 25, 32, and 38 GHz (Fig. 1), whose lengths correspond to the length scales of the catchment and can, therefore, capture rainfall spatial variability at sub-kilometer scale. The selected CMLs are standard duplex links operated on MINI-LINK Ericsson platform with ATPC configuration (switched on during the whole experimental period).

"Ground truth" rainfall observations are collected at four locations by six tipping bucket RGs (MR3, Meteoservis v. o. s.,

Czech Republic), two of them are collocated (Fig. 1, left). Each RG is dynamically calibrated (once a year), and checked and maintained at least once a month. In addition, six RGs from the operational rainfall monitoring network of the municipality are used (Fig. 1, middle) to test the effect of RG layout on CML adjusting efficiency. These RGs are also dynamically calibrated (Stransky et al., 2007). All RGs are the same type with a catch area of 500 cm$^2$ and a quantization of 0.1 mm.

### 2.2 Estimating areal rainfall from three different rain gauge layouts

To investigate in how far limited representativeness of point RG observations together with space-time rainfall peak averaging affects the performance of adjusted CMLs, we estimate the areal rainfall from three different RG layouts A, B1 and B2 (Fig. 1, left). The Layout A is a single RG located inside the catchment. This is a typical configuration used by engineering companies when calibrating rainfall-runoff models of small urban catchments. Layouts B1 and B2 consist of three RGs located outside the catchment.

In B1, RGs are relatively close to the catchment. They form a triangle with edge lengths of 7.0 km, 5.4 km and 2.8 km with the catchment area approximately in its center. In B2, the RGs are more distant and form a triangle with edges 11.5 km, 9.6 km and 8.2 km with the catchment closer to the NE vertices (Fig. 1, left).

### 2.3 Experimental periods

The experimental period in the Dübendorf site was between June 2012 and September 2012. During this period, 19 events

exceeded 5 mm in total and thus are relevant for stormwater management (Table 1). The experimental period for the Prague-Letnany site was between June 2014 and October 2014. During this period, 13 relevant events occurred (Table 1).



## 2.4 Simplified attenuation-rainfall model

To adjust CML continuously in near real-time, equation (1) is especially by shorter links (less than 1÷2 km) not suitable because signal-to-noise ratio of CMLs is often low and power-law retrieval model (1) is prone to overfitting.

We, therefore, propose a simple two-parameter linear attenuation-rainfall model which is intended for commercial CMLs

between approx. 20–40 GHz, i.e. frequencies frequently used by mobile network operators for shorter hops in urban areas:

$$R = \gamma(k - k_w) \qquad (2)$$

where $\gamma$ [mm h$^{-1}$ km dB$^{-1}$] is an empirical parameter, $k$ [dB km$^{-1}$] is a specific attenuation after baseline separation and $k_w$ [dB km$^{-1}$] is an offset parameter which corrects for wet antenna attenuation and possible bias introduced by inaccurate baseline identification. The linearity of the relation makes it possible to condition the model to rainfall and attenuation data

which were aggregated over relatively long intervals (e.g., hours) and at the same time predict rainfall for attenuation data sampled at high frequencies.

## 2.5 Conditioning the simplified attenuation-rainfall model

First, RG rainfall intensities and CML attenuations are averaged to the same time resolution and appropriate aggregation intervals. The rainfall-attenuation model is then continuously fitted on aggregated data using moving window of $N$

consecutive data points, i.e. for each time step $i$ one set of model parameters ($\gamma$, $k_w$) is identified. Only data points with non-zero rainfall are included into the calibration window as the model is designed for wet weather periods. We tested different window lengths ($N = 3, 5, 10$ points) and found that the optimal $N$ in our case is five points (see section 4.1. for more details). In general, longer window (larger $N$) reduces sensitivity to the random noise but requires stronger stationarity of error models.

The model (2) is fitted by minimizing cost function $L$ using a gradient method based on a quasi-Newton optimization algorithm L-BFGS-B implemented in the R language function optim()(Byrd et al., 1995):

$$L = \sum_{i-N+1}^{i}\left(\hat{R}_i - \tilde{R}_i\right)^2, \qquad (3)$$

where, $\hat{R}$ is observed aggregated RG rainfall and $\tilde{R}$ is rainfall produced by model (2). In this study, we carried out two consecutive optimization runs for each attenuation-rainfall time series. First optimization run (a) is implemented with wide

parameter ranges and the second run (b) is performed with parameters constrained based on previous model realizations. For the first optimization run (a), lower limits of both parameters are set to zero. This avoids negative parameter values which do not have a physical meaning. The upper limit of the parameter $\gamma$ is set to the ITU recommended value for parameter $\alpha$ in Eq. (1) (ITU, 2005) increased by 50 % to compensate for the effect of exponent $\beta$ in Eq. (1) during heavy rainfalls. The upper limit of the parameter $k_w$ is set proportionally to the inverse of CML length (5 dB km$^{-1}$), which corresponds approximately to

wet antenna attenuation offsets reported by Leijnse et al., (2008).

New parameter ranges for optimization run (b) are estimated from parameter distribution of run (a): i) parameter values settled at upper limit are removed, as these are likely to be associated with outliers, ii) only parameters associated with





a specific attenuation k > 1 dB km$^{-1}$ are considered, iii) new parameter ranges are set from the remaining values as 5 % and 95 % quantiles.

## 2.6 Performance assessment

The performance of the method is evaluated by direct comparison of CML QPEs with reference rainfall both having a one-minute resolution. CML QPEs are evaluated for rainfall events listed in Table 1. In addition, results are compared with unadjusted CMLs processed by standard models with fixed parameters.

The effect of rainfall aggregation on CML adjusting is investigated on four CMLs from T-Mobile's network in Prague-Letnany (CZ) and one commercial CML operated for experimental purposes in Dübendorf (CH). In this investigation, RGs used for CML adjustment are also reference RGs against which CMLs are evaluated. The only difference between rainfall for adjusting and reference rainfall is time resolution. Aggregation intervals from five minute to one day (5 min, 15 min, 30 min, 60 min, 3 h, 6 h, 12 h, and 1 d) are used for CML adjustment, whereas the performance is evaluated on one-minute data. The influence of RG layout on CML adjusting is tested on data from Prague only.

The QPEs from unadjusted CMLs are calculated using a standard power-law model (1) and wet antenna corrections with fixed parameters. The Prague CMLs are corrected for wet antenna attenuation using the constant correction as suggested by Overeem et al. (2011). The Dübendorf CML is corrected for wet antenna attenuation by a specific model suggested by Schleiss et al. (2013). Both power-law and wet antenna attenuation models are applied under two scenarios: S1) with parameters from literature (ITU, 2005; Overeem et al., 2011; Schleiss et al., 2013) and S2) with local parameters inferred from the available reference data.

The performance of the algorithms is then evaluated for each event. First, the coefficient of determination ($R^2$) is used to evaluate the ability of CMLs to capture rainfall temporal dynamics. $R^2$ is a relative measure which gives comparable results of CML performance even for events of different characteristics. Second, the systematic deviations of CMLs are assessed by plotting their QPEs against reference RGs and evaluated quantitatively by the slope of a linear regression (linear trendline intersect set to zero). In addition, the relative error in cumulative rainfall is calculated for each single event as the relative difference between the QPEs and reference rainfall amounts.

## 3 Results

First, the performance of CMLs when adjusted with rainfall of different time resolution is presented. Both results from Dübendorf (CH) and Prague-Letnany (CZ) are shown (Fig. 2 and 3). Second, the influence of different RG layouts on CML adjusting is demonstrated on Prague's dataset (Fig. 4 and 5). Finally, QPEs from adjusted CMLs are compared with the application of standard attenuation-rainfall models (Fig. 6). The CML performance is in all three cases evaluated on data with one minute temporal resolution.



### 3.1 Influence of different aggregation intervals

The performance of CMLs adjusted by rainfalls aggregated to 5 min, 15 min, 30 min, 60 min, 3 h, 6 h, 12 h, and 1 d intervals is presented below. Relative error in cumulative rainfalls and $R^2$ is shown for each link and aggregation (Fig. 2). In addition, for Prague-Letnany, the mean QPEs from all CMLs are evaluated.

It can be seen that the continuous adjustment performs well for aggregation intervals up to one hour (rel. error < 7 %, $R^2 > 75$ %). CML QPEs adjusted to (sub)hourly data are associated with low systematic errors and reliable rainfall intensities over the whole range from light to heavy rainfall (Fig. 2). We only find a slight underestimation of high intense peaks (Fig. 3), which might be due to mismatch between point and path-averaged observations. The best performance is achieved when the QPEs from all CMLs are averaged. This is probably due to reduction of random errors, when nearly unbiased rainfall

information from multiple sensors is merged. In addition, multiple CMLs cover the catchment area better than a single CML. The performance of the adjustment algorithm substantially decreases when aggregation interval is increased from 1 h to 3 h and then further to 6 h and 12 h (Fig. 2, $R^2$). This is probably associated with the extent to which rainfall autocorrelation characteristics are preserved when aggregating rainfall data to coarser time resolution (Appendix A, Fig. A1). Hourly aggregations still seem to correspond relatively well to the temporal scale of rainfall peaks, whereas three-hour sums already

often smoothes out peak intensities by averaging them over periods with low-intensity or zero rainfall. This averaging probably impacts the identifiability of the parameters of the simplified model (2).

When evaluating systematic errors for each event separately its variability is increasing with increasing aggregation interval up to 12 h. Surprisingly, adjusting CMLs to daily rainfall volumes leads to less variable results, although more biased in average (Fig. 2 and 3). This might be caused by the correlation structure of rainfall, where the correlation between peak

intensities is better preserved by daily than 12 hours aggregations (Appendix A, Fig. A1). In addition, fluctuations of CML baseline have, according to our experience, a daily pattern and thus rainfall with daily resolution can be appropriate to minimize the effect of these fluctuations.

For the Dübendorf data, the method also does not perform well for long aggregation intervals > 1 h (Fig. 2 and 3). However, here the mismatch most probably stems from the different effect: antenna wetting attenuates the transmitted signal for up to

six hours after rainfall has stopped (Fig. 2 in Schleiss et al., (2013)). Aggregating these dry weather periods with increased attenuation over longer time intervals then causes substantial error in adjusted QPEs, because this process is not considered in the simplified model. Interestingly, we find that the drying times of CMLs from Prague-Letnany are considerably shorter, mostly within few minutes. The reasons for this effect are not known.

### 3.2 Influence of different rain gauge layouts

The performance of the algorithm for different RG layouts is evaluated on the Prague-Letnany dataset. For each layout, the rainfall was aggregated to 5 min, 15 min, 30 min, and 1 h time resolution. We found that the best performance was achieved by averaging all four short CMLs located in the catchment - for all RG layouts. The performance of single CMLs is slightly





worse. The relative differences between QPEs from single CMLs and from their averages are in very similar proportions by all CMLs as when adjusting to reference rainfall (see the previous section). Therefore, only the performance of averaged QPEs from all four CMLs is presented.

**Layout A:** CMLs adjusted by the single RG located in the catchment measure very well both light and heavy rainfalls - with the exception of slight underestimation of high-intense peaks over 30 mm h$^{-1}$ (Fig. 4). The median systematic error of CML QPEs corresponds to the bias of the single RG (Fig. 5). Nevertheless, adjusted CMLs clearly outperform a single RG in terms of capturing rainfall temporal dynamics. The median $R^2$ of CMLs is between 0.85 and 0.87 where the highest $R^2$ is (0.77–0.94) is obtained for an aggregation interval of 15 min. The inter-event variability of $R^2$ slightly increases for longer aggregation intervals reaching values 0.70–0.90 for 1 h. These are much higher values of $R^2$ than those reached by the RG layout A alone, 0.52– 0.78 with median 0.68 (Fig. 5).

**Layout B1:** CMLs adjusted to three rain gauges close to the catchment perform slightly worse than CMLs adjusted by the layout A. In Fig. 4, a systematic underestimation of intense rainfalls is visible. It is most pronounced for intensities exceeding 30 mm h$^{-1}$ and, in contrast, light rainfalls are overestimated by the CMLs. The bias in RG areal rainfall used for adjusting (evaluated for each event separately) varies substantially more than the one from the layout A. This also leads to a higher variability in the systematic error of QPEs. Interestingly, $R^2$ for the CMLs (Fig. 5) is only slightly lower (median is between 0.80 and 0.84) than for CMLs adjusted by the layout A, but has a higher variability. The best performance is achieved for 15-min aggregation interval with the narrowest range of relative errors in cumulative rainfalls (-0.32–0.25) and a $R^2$ (0.68– 0.94).

**Layout B2:** We find that CML which are adjusted to three distant rain gauges reliably capture light and moderate rainfalls but substantially underestimate heavy rainfall peaks (Fig. 4). Systematic errors and inter-event variability are only slightly higher than for layout B1. As expected, for the distant gauges the best performance in terms of $R^2$ value and its variability is achieved for longer aggregation intervals. The $R^2$ for adjustment with hourly aggregation intervals ranges between 0.50–0.91 with median 0.78. The poor performance for 5 min aggregation intervals (low values of $R^2$) can be explained with both the underestimation of high intense rainfall peaks and errors in the "ground truth", because at the spatial scale of RG layout B2 aggregation interval of 5 min is insufficient to average out discrepancies between point and areal rainfall intensity.

In summary, the optimal aggregation interval to adjust CMLs for a given catchment and RG layout increases with larger RG-CML and RG-RG distances. This is, because time aggregation, in general, improves the spatial representativeness of point RG measurements (Villarini et al., 2008). However, computing areal rainfalls over increasingly large area also increasingly smoothes out rainfall peaks, which propagates also to CML adjusted QPEs. Therefore, CMLs adjusted to relatively distant RGs perform the worst in comparison with the other RG layouts. Considering the performance of RGs alone, the benefit of using the RGs in combination with CMLs is clearly visible (Fig. 4 and 5) even in the case of layout B2 with RGs relatively distant from the catchment. Although we can demonstrate the effect of peak averaging with our experimental data, further research is needed to adjust CMLs to remote RGs while preserving peak rainfall intensities.



### 3.3 QPEs from unadjusted CMLs

To demonstrate the need for an effective adjustment procedure, standard k-R power-law (1) and wet antenna attenuation models with fixed parameters were used to retrieve QPEs from unadjusted CMLs according to the state-of-the-art (Overeem et al., 2011; Schleiss et al., 2013). The results are presented for two simulation scenarios S1) model parameters taken from literature (ITU, 2005; Overeem et al., 2011; Schleiss et al., 2013), and S2) parameters obtained by fitting models to the reference dataset.

First, the results for scenario S1 show a positive bias for the QPEs from Prague-Letnany, which on average is about 50 %. This bias leads to the unsatisfactory performance of single CMLs also in terms of $R^2$. The averaging of observations cannot compensate for this bias and thus cannot substantially improve the $R^2$, which measures the reliability of the retrieval model. Second, the QPEs from the Dübendorf CML are much more reliable both in terms of smaller systematic deviations and a large $R^2$. In addition, variability is low, which means that it performs well even for very light and extremely heavy events. This is due to the extremely good reference data, which made it possible to tailor a custom model for wet antenna attenuation correction for this particular CML (Schleiss et al., 2013).

For scenario S2, model fitting leads to substantial reduction of bias in Prague-Letnany CML observations, in contrast to that, the bias of the Dübendorf CML remains almost unchanged. This reduction leads to a much better $R^2$. The best performance in terms of $R^2$ is achieved for QPEs calculated as a mean from all Prague-Letnany CMLs. The $R^2$ of Dübendorf CML is comparable to the value when scenario S1 was used (Fig. 6).

The unadjusted QPEs from Prague CMLs in scenario S1 are substantially less reliable than QPEs from any adjusted CML presented above (Fig. 2, 4, and 6). The performance of Prague CMLs treated with models with optimal parameters (S2) corresponds approximately to the CMLs adjusted with three hours cumulative rainfalls (Fig. 2) or adjusted by RG layout B2 (Fig. 4). The performance of unadjusted Dübendorf CML (for both scenarios) corresponds, similarly as in Prague-Letnany, to adjustment to an aggregation interval of 3h (Fig. 2).

The relatively bad performance of unadjusted Prague-Letnany CMLs under scenario S1 compared to Dübendorf CML is partly caused by their short paths (1020 m, 650 m, 1400 m, and 610 m, compared to 1850 m). In addition, the automatic power control, which was switched off for the Dübendorf CML, also reduced the performance. We found that automatic power control worsens the quantization of CMLs (as $Tx$ has about three times lower quantization than $Rx$) and thus one can learn less from observations about the parameters of the retrieval models, especially from short CMLs. An automatic power control as a standard feature of today's CMLs needs to be considered when modern CML networks are used for rainfall monitoring. The results, however, indicate, that combining rainfall observations from multiple unbiased (or slightly biased) CMLs reduces such random errors by averaging and thus improves QPEs for areal rainfall.





## 4 Discussion

The goal of this study was to suggest a procedure to adjust QPEs from CMLs to local rain gauges and to demonstrate the benefits over current retrieval methods. We obtained very promising results, with relative errors of a few percent. Although these are promising results we would like to discuss, first, errors associated with the linear approximation of attenuation-rainfall model Eq. (2) and WAA models and, second, how to condition the model (2) to local RG observations. Third, we would like to discuss limits of the proposed adjusting algorithm, e.g. regarding preservation of peak rainfalls.

### 4.1 Linear approximation of the power-law retrieval model

The model (2) can be interpreted as a combination of linear forms of attenuation-rainfall model (1) and WAA models. The uncertainty due to the simpler model structure of Eq. (2) is comparable especially for shorter links with quantization of CML readings. To illustrate this effect, we compare the results for Eq. (1) and Eq. (2) by predicting specific attenuations for rainfall intensities from 0 to 60 mm h$^{-1}$. The power-law model uses the ITU parameters (ITU, 2005), the linear model is fitted to the results of the power-law model by minimizing maximal absolute deviation. In Fig. 7, the results for 38 GHz CML are shown, because the deviations for 38 GHz are larger than for 25 and 32 GHz due to the relatively high value of exponent beta (1.13) for vertically polarized 38 GHz CML. The deviation between the power-law model and simplified model are between ± 1.5 mm h$^{-1}$, which corresponds to a specific attenuation of approx. 0.5 dB km$^{-1}$. The deviation between WAA models and appropriate linear approximations depend on their character. E.g. the WAA model of Overeem et al., (2011) is only based on single additive parameter and is thus fully included in our model through the parameter $k_w$. Interestingly, a linear approximation of the coupled attenuation-rainfall model (1) and Kharadly's WAA model (Kharadly and Ross, 2001) leads to considerably higher deviation (Fig. 7, Middle). The deviation can be, however, substantially reduced by fitting the linear model over a narrow range of attenuations, resp. rainfall intensities. For example, the right panel of the Fig. 7 shows two linear models fitted separately for lighter (R <= 12 mm h$^{-1}$) and heavier rains (R > 12 mm h$^{-1}$). The absolute deviation between the linear approximations and the original model is less than one third compared to linear fit over a whole range of rainfall intensities (Fig. 7, middle).

When fitting the simplified model Eq. (2) continuously over relatively short periods, it is likely that the rainfall intensities covered by the calibration window will vary in narrow ranges resulting in relatively small errors introduced by linear approximation. However, although the length of the calibration window reduces the effect of random errors, its optimal length also depends on the stationarity of CML errors which in turn depend on characteristics of the rain event, the CML hardware and the local environment (see introduction). For both experimental sites, we identified the window length of five points as an acceptable compromise between window length and the temporal variability of rainfall.





## 4.2 Attenuation-rainfall model fitting

The time aggregation of rainfall and attenuation data smoothes out rainfall peaks. This leads to narrower intervals of likely parameter values and especially lowers the upper bound of resulting parameter estimates. As an example, the resulting parameter distributions are shown here for the CML 2 (Prague-Letnany) when adjusted to rainfalls for different aggregation intervals (Fig. 8). The peak averaging reduces the width of the parameters distribution and thus limits the ability of the model to predict high rainfall-intensities, which are mostly associated with large values of $\gamma$. A similar tendency can be seen for spatial averaging when CMLs are adjusted based on areal rainfall estimated from RGs which cover a larger region.

The substantial difference in values of $\gamma$ parameter (Fig. 8) compared to parameter $\alpha$ of the model (1) suggested by ITU (ITU, 2005) is caused by the conceptual difference between models: Proposed model (2) is a combination of wet antenna attenuation model and simplified standard power-law model. Such discrepancy was already reported by Fenicia et al. (2012), who estimated for their 23 GHz link values of $\alpha$ substantially lower than values suggested by ITU.

## 4.3 Limits of the proposed adjustment method

In our study, we focus on urban rainfall monitoring and adjust CMLs with path lengths fewer than 2 km. For these CMLs, adjusting to "ground truth" measurements with aggregation intervals up to one hour is accurate and only slightly underestimates high intense rainfall peaks. The use of rainfalls with longer aggregation intervals, e.g. from 3 h to 1 d, however, leads to systematic underestimation of high intense rainfalls and slight overestimation of low intense rainfalls (Fig. 2 and 3). We have found a connection between this systematic discrepancy and the extent to which rainfall autocorrelation is preserved in the aggregated rainfall (Fig. A1). Nevertheless, further research is needed to develop a method which would correct these systematic errors based on the spatio-temporal correlation of rainfalls in the region of interest.

The performance of the proposed adjustment method is also dependent on the spatial layout of the "ground truth" measurements. The spatial averaging, similarly as time averaging, smoothes out rainfall extremes, i.e. layouts where the RGs are further away from the CML or each other, tend to underestimate rainfall peaks. Even worse, these larger distances cause bias in the "ground truth" observations, because the probability increases that distant gauges completely miss (or hit) actual peak intensities. The optimal aggregation interval for layout B2 was 1 h, whereas the optimal interval for A and B1 was only 15 min. This is, because longer time averaging reduces discrepancies between areal and point rainfall estimates. The factor to which high intense rainfalls are systematically underestimated corresponds quite well with the areal reduction factor reported in literature (e.g., Hydraulics Research, 1983). This indicates that the systematic underestimations associated with areal averaging might be reduced based on climate-specific rainfall characteristics. An interesting idea is to directly infer the spatio-temporal variability of a certain rain event from the observations of many CMLs. However, further research is needed to incorporate these features into an improved adjustment procedure.

Last, but not least, the reliability of the adjustment corresponds to the reliability of the "ground truth" observations. One possibility to ensure good reference data could be to use CMLs to eliminate gross errors, e.g. by identifying malfunctioning



RGs (Bianchi et al., 2013b) and excluding them from CML adjustment. Another possibility, which should be investigated in the future, is to use longer CMLs of appropriate frequencies instead of RGs in the adjustment. As argued in section 1.2, these long CMLs are less sensitive to hardware and environmental influence factors. Nevertheless, our personal experience after working several years with signal attenuation from many operational CMLs is that it happens rather often that CML data

show erratic and seemingly random behavior and that the response to rainfall does not always correspond to a power-law relationship. While we at this time can only speculate about the reasons, it is crucial to carefully select and test those long CMLs which should serve as a reference.

## 5 Conclusions

Commercial microwave links (CMLs) can measure rainfall in sparsely gauged regions and improve the resolution of existing

rain gauge and radar networks, especially in populated areas where they are often very dense. Quantitative precipitation estimates (QPEs) from CMLs as rainfall sensors are, however, affected by various uncertainties, which are still too poorly understood to build effective signal-processing algorithms based on CML observations alone. In this paper, we therefore suggest a generic method to adjust CML QPEs to aggregated observations from existing RGs such as 15min or hourly averages:

● Our results demonstrate that standard commercial CMLs operated by mobile network operators can be used as powerful sensors for capturing rainfall variability at (sub)minute scale. Combining the high-resolution observations from CMLs with the reliable cumulative observations from RGs enables us to derive reliable QPEs of high temporal resolution and very good spatial representativeness. Thus, our method can also be seen as a method for spatio-temporal disaggregation of cumulative RG measurements based on CML attenuation.

● We propose a simplified semi-empirical model for CML rainfall estimation which combines microwave attenuation from rain and antenna wetting into one linear relation. The model can be easily continuously adjusted to rainfall from existing RG networks in operational conditions, even though RGs may have a low spatial coverage and temporal resolution. The model is intended for short CMLs (path length ≈ 1÷2 km or less) operating at frequencies approx. between 20-40 GHz, where the model structure errors from the linearization are much smaller

than other influence factors, such as for example the quantization of CML attenuation. These CMLs are crucial for capturing rainfall space-time structure at the fine scale required for urban hydrological applications.

● Our simple and robust approach performs very well for CMLs adjusted by rainfall with aggregation intervals up to one hour. Adjusting CMLs with longer aggregation intervals, however, leads to systematic underestimation of high intense rainfalls and slight overestimation of low intense rainfalls. We have found

a connection between this systematic discrepancy and a degree to which autocorrelation structure is preserved in aggregated rainfall data.



● We have demonstrated on three different RG layouts that the CMLs adjusted by the RGs provide substantially better areal QPEs than the RGs alone. However, RG layouts which cover larger areas, e.g. approx. $10÷100$ km$^2$, tend to underestimate rainfall peaks and slightly overestimate light rainfalls, which is similar to the effect observed by temporal averaging. We have found that the underestimation is proportional to the areal reduction factor reported in the literature.

● Further research towards an improved adjustment method which reduces systematic discrepancies in adjusted CML QPEs by explicitly considering space-time characteristics of rainfalls seems very promising. The rainfall space-time structure might be incorporated in the model by correction factors based on either local climatology or by directly estimating it from the response of the CML network itself. The latter seems especially interesting for ungauged regions, where longer CMLs might provide reliable reference rainfall to correct shorter CMLs.

The proposed approach overcomes one of the biggest shortcomings of commercial CMLs as rainfall sensors for practical use in urban hydrological application: the calibration of CML rainfall estimation models to site-specific conditions.

The adjustment of CMLs to cumulative rainfall from point ground measurements has a huge potential especially for urban catchments, where the CML network is commonly very dense. The combined use of RGs and CMLs can thus greatly improve the spatial and temporal resolution of existing rainfall products and contribute to better understanding urban rainfall runoff processes, which are often hampered by poor rainfall data. Moreover, the insight into rainfall space-time structures at (sub)minute and (sub)kilometer resolution can contribute to deeper understanding of rainfall behavior at the microscale.

**Appendix A: Temporal rainfall aggregation**

Aggregating rainfall over time reduces the discrepancies between point, path-averaged, and areal rainfall, but also smoothes out rainfall dynamics (Villarini et al., 2008) which would make it possible to better identify attenuation-rainfall model parameters. The effect of rainfall intensity averaging when increasing the aggregation interval is demonstrated on the rainfall data from our reference RGs in Prague-Letnany (CZ). The original rainfall time series with one-minute resolution (Fig. A1, top row) is aggregated over eight different integration times from five minutes (second row) to one day (bottom row). The resulting time series are compared with the original one. Only periods belonging to events listed in the Table 1 are selected, which restricts the analysis only to rainy periods with significant intensities. The right panel of Fig. A1 shows the correlation between entire time series (blue) and the correlation between rainfall intensity maxima of each event (red). It can be seen that the temporal aggregation up to one-hour preserves the main characteristics of rain events in Prague very well, e.g. high-intensity convective rainfalls can be recognized from low-intensity frontal rainfalls.





## Competing interests

The authors declare that they have no conflict of interest.

## Acknowledgements

This work was supported by the project of Czech Science Foundation (GACR) No. 14-22978S, the project of the Czech
Technical University in Prague project No. SGS16/057/OHK1/1T/11, and the Swiss National Science Foundation in the
scope of the COMCORDE project No. CR2212_135551. We would like to thank T-Mobile Czech Republic a.s. for kindly
providing us CML data and specifically to Pavel Kubík, for being helpful with our numerous requests. Special thanks belong
to Prazske vodovody a kanalizace, a.s. who provided and carefully maintained the rain gauges. Last but not least we would
like to thank Eawag for supporting COMMON project and to Prazska vodohospodarska spolecnost a.s. for providing us
additional rainfall information from their RG network.

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





Table 1 – Rainfall events selected for the evaluation at Prague-Letnany site, CZ in 2014 and Dübendorf site, CH in 2011. The maximal intensity *R_max* and the total rainfall amount *H* are provided for each event. Short convective rainfalls with peak intensities up 90 mm h$^{-1}$ and long low-intense stratiform rainfalls are included in the datasets.

| Prague-Letnany, CZ | | | | Dübendorf, CH | | | |
|---|---|---|---|---|---|---|---|
| Beginning (2014) | Duration [min] | R_max [mm h$^{-1}$] | H [mm] | Beginning (2011) | Duration [min] | R_max [mm h$^{-1}$] | H [mm] |
| 21 Jul 15:01:00 | 600 | 19.1 | 13.7 | 13 Jul 13:55:00 | 330 | 7.7 | 14.0 |
| 11 Aug 01:01:00 | 780 | 5.6 | 7.7 | 17 Jul 06:30:00 | 620 | 5.5 | 9.2 |
| 14 Aug 14:01:00 | 180 | 38.7 | 5.0 | 19 Jul 13:55:00 | 430 | 5.5 | 10.5 |
| 16 Aug 13:01:00 | 180 | 24.5 | 5.3 | 23 Jul 23:30:00 | 225 | 11.2 | 8.2 |
| 26 Aug 21:01:00 | 720 | 5.2 | 8.7 | 27 Jul 13:30:00 | 90 | 24.0 | 5.2 |
| 01 Sep 13:01:00 | 1200 | 2.7 | 12.9 | 27 Jul 17:20:00 | 125 | 22.7 | 5.9 |
| 11 Sep 13:01:00 | 1560 | 59.7 | 40.5 | 05 Aug 18:00:00 | 150 | 76.5 | 18.7 |
| 14 Sep 16:01:00 | 240 | 13.5 | 7.3 | 07 Aug 05:40:00 | 165 | 14.4 | 5.7 |
| 21 Sep 19:01:00 | 420 | 8.6 | 7.3 | 14 Aug 23:25:00 | 290 | 19.0 | 9.2 |
| 13 Oct 22:01:00 | 600 | 18.2 | 18.1 | 15 Aug 11:00:00 | 140 | 92.4 | 20.6 |
| 16 Oct 03:01:00 | 420 | 22.7 | 6.6 | 24 Aug 16:50:00 | 280 | 9.9 | 10.9 |
| 21 Oct 21:01:00 | 300 | 11.4 | 6.3 | 26 Aug 23:40:00 | 305 | 8.9 | 12.7 |
| 22 Oct 10:01:00 | 420 | 4.8 | 6.5 | 01 Sep 03:10:00 | 110 | 54.0 | 5.9 |
| | | | | 03 Sep 19:00:00 | 220 | 75.4 | 9.8 |
| | | | | 04 Sep 14:40:00 | 360 | 18.2 | 16.3 |
| | | | | 04 Sep 22:15:00 | 245 | 17.7 | 5.8 |
| | | | | 14 Sep 02:25:00 | 275 | 13.8 | 8.5 |



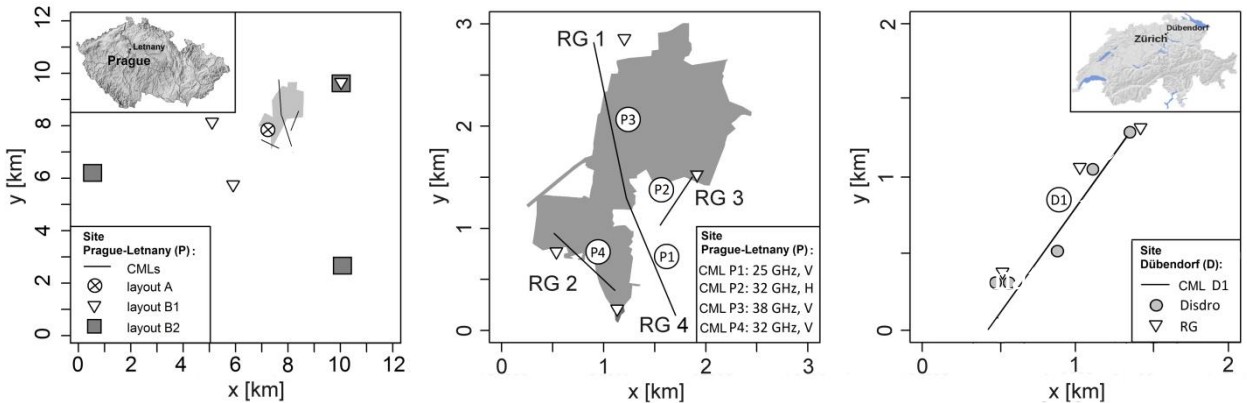

**Figure 1: Experimental sites Prague-Letnany, CZ (left and middle) and Dübendorf, CH (right). Left: Overview CZ, RG layouts used for CML adjusting. Middle: Detailed view on CZ, CMLs and reference RGs. Right: Detailed view on CH, CML and the layout of reference disdrometers and RGs.**

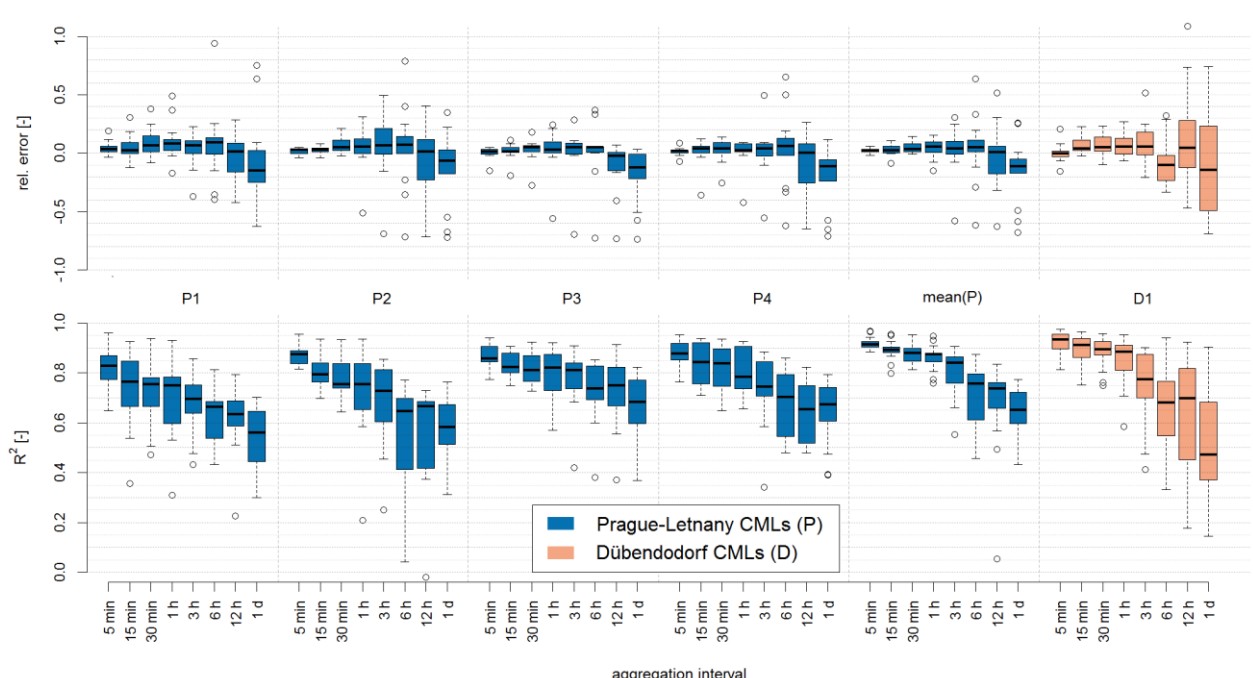

**Figure 2: Relative error (top) and $R^2$ (bottom) in QPEs of CMLs adjusted by rainfall data of different time resolution. Each CML layout is represented by eight boxplots corresponding to QPEs adjusted by rainfall aggregated to time intervals from 5 min to 1 d. Each boxplot depicts a range of the statistics during all evaluated events. Five groups of blue boxplots (left) evaluate QPEs from**
10 **single CMLs and from their average at Prague-Letnany. One group of orange boxplots (right) depicts QPEs from a single CML at Dübendorf.**





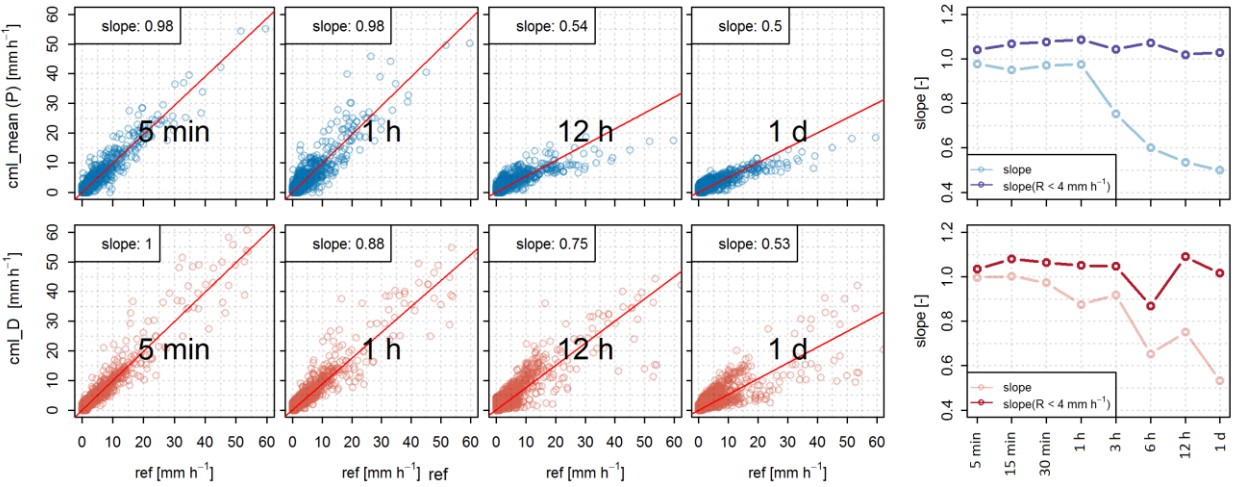

**Figure 3:** Comparison of CML QPEs adjusted by rainfall data of different time resolution to reference rainfall, from four averaged CMLs in Prague-Letnany (top) and one CML in Dübendorf (bottom). Scatter plots are shown only for selected aggregation intervals. Linear trendline intersects are set to zero. Slopes of trendlines for all aggregation intervals are depicted in the right panels, showing also slopes of trendlines calculated for light rainfalls (R < 4 mm h$^{-1}$).

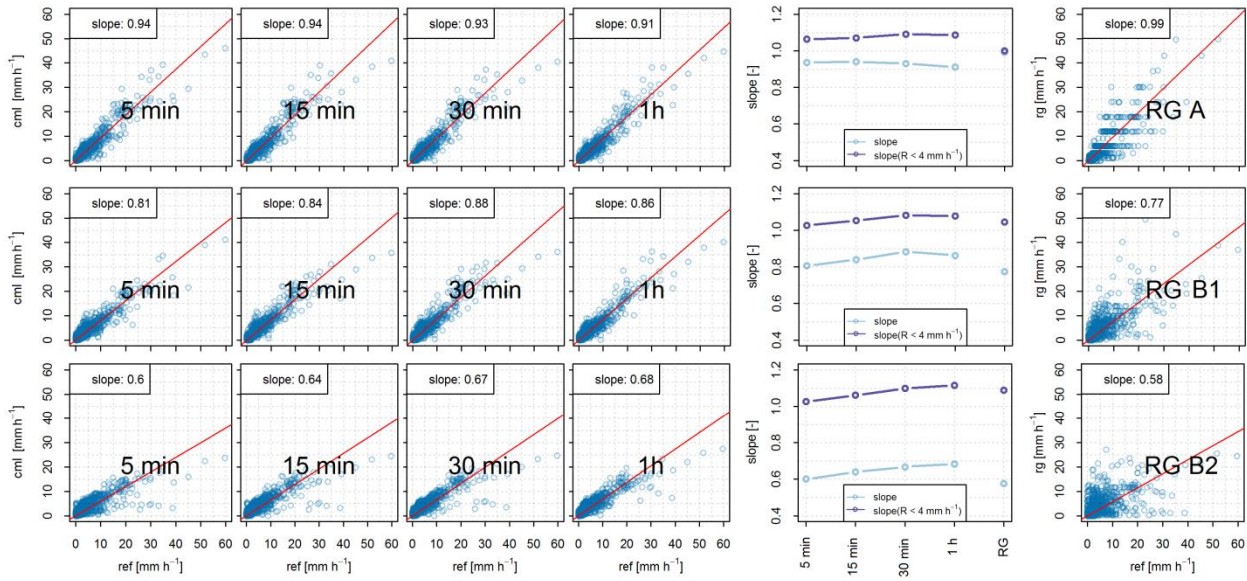

**Figure 4:** Comparison of mean QPEs from four CMLs to reference rainfall. CMLs are adjusted by rainfall from three different RG layouts (rows) with aggregation intervals of 5 min, 15 min, 30 min, and 1 h (four panels on the left), in addition, rainfall from the RG layouts alone is compared to the reference areal rainfall (right panel). Linear trendline intersects are set to zero. The middle panel plots the relationship between the slope of the trendlines and aggregation times as well as the slope of the RG layouts.



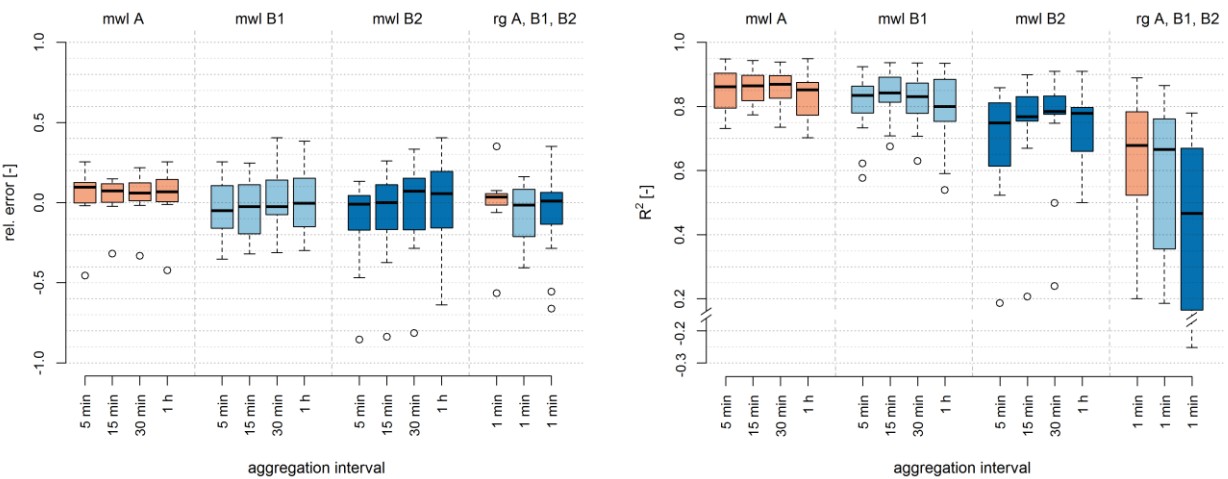

**Figure 5: Relative error (left) and $R^2$ (right) in QPEs from CMLs when adjusted using three different RG layouts (A, B1, B2) and four different aggregation intervals (5 min, 15 min, 30 min, and 1 h). Right three boxplots in both figures correspond to RG observations of each layout when used alone without CMLs.**

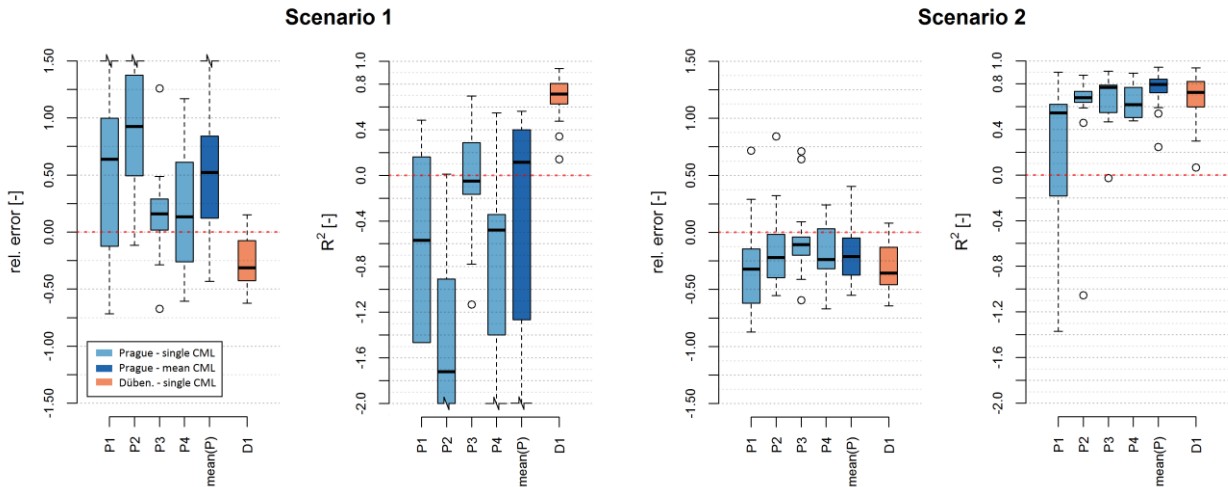

**Figure 6: Relative error and $R^2$ in QPEs of unadjusted CMLs evaluated for all events. Each boxplot depicts one CML (resp. CML mean). Scenario 1: QPEs based on models with parameters from the literature. Scenario 2: QPEs based on models with optimal parameters. It can be seen that choosing parameter values for the retrieval model from literature leads to large positive bias**
10 **(scenario 1, rel. error). Conditioning the model on observations leads to a negative bias, albeit with reduced variance. Both do not achieve the virtually unbiased observations obtained with our adjustment method, with are an order of magnitude lower (Fig. 2). The comparably good performance of the CML D1 is due to an exceptional ground truth which enabled a custom-made wet antenna correction.**





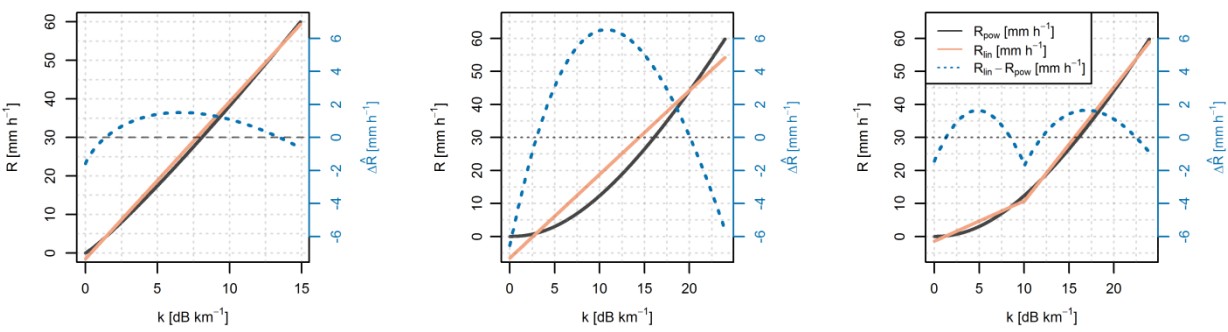

**Figure 7: Performance of linear approximation of k-R models for vertically polarized 38 GHz CML in terms of rainfall intensity. Left: Linear approximation (red) of the power-law model (black). The blue dashed line shows the resulting model structure errors. Middle: Linear approximation of power-law model coupled with Kharadly's wet antenna attenuation model. Right: Power-law model combined with Kharadly's wet antenna attenuation model approximated by two linear models fitted separately for light (R <= 12 mm h⁻¹) and heavy rainfall events (R > 12 mm h⁻¹).**

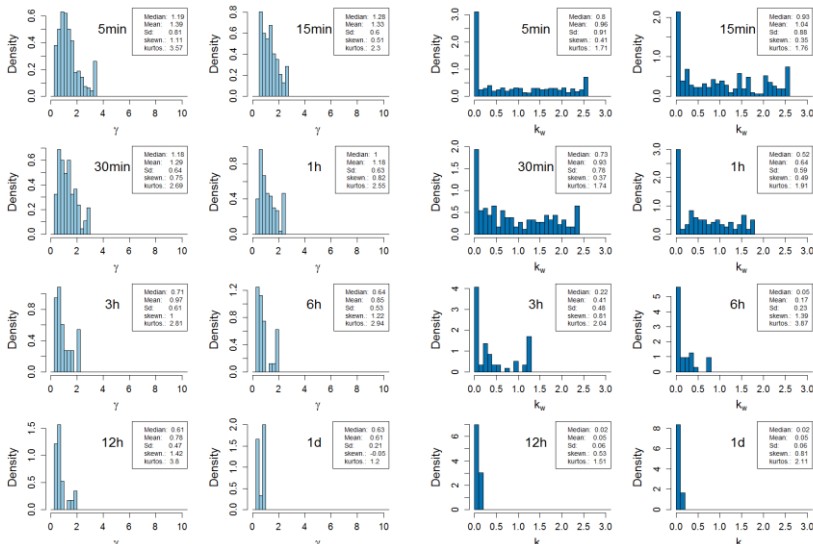

**Figure 8: Parameters $\gamma$ and $k_w$ of the model (2) fitted for the CML 2 (32 GHz, horizontally polarized) using rainfall data of different aggregation intervals. Each histogram corresponds to the distribution of one parameter optimized on data of a given aggregation interval. Only parameters associated with model realizations with a specific attenuation larger than 1 dB km⁻¹ are depicted by the histograms.**





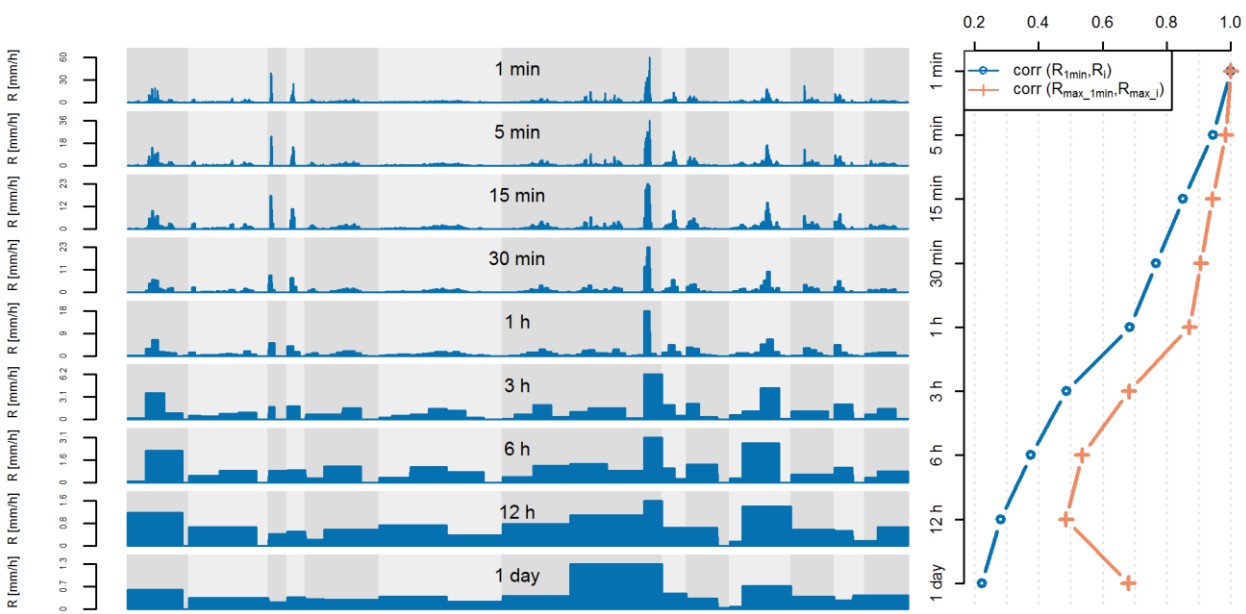

**Figure A1: Rainfall peaks smoothed out by longer aggregation intervals, here shown for the case study in Prague (CZ). Left: Merged time series of thirteen events aggregated to time steps from 1 min to 1 d. Vertical stripes indicate individual events. Note how the range of the y-axis decreases from the top to the bottom row. Right: Correlation between time series with 1 min resolution and the other time series of different resolutions (blue) and correlation between peak intensities of events derived from rainfall data with 1 min resolution and peak intensities derived from aggregated rainfall data (red).**