# Peer review of "Gauge-Adjusted Rainfall Estimates from Commercial Microwave Links"

_Hydrology and Earth System Sciences, 2016_

## Referee Comment (RC1) · H. Leijnse (Referee) · 27 Sep 2016

This paper describes a method for incorporating accurate rain gauge measurements in commercial microwave link (CML) rainfall estimation through on-line parameter adjustment of the CML retrieval model. The idea of adjusting those model parameters that we know are most uncertain based on rain gauges is very appealing. This means that the accuracy of the gauges is used where it is most needed. The authors test their method on two different datasets, with different algorithm settings and different distances to the gauges used for adjustment. I think that the paper is interesting and certainly appropriate for HESS. I also have some issues that I think the authors should deal with before the paper is ready for publication. The most important of these issues are: 1) How well does the presented method work when gauges are even further away from the links (i.e., how well can this method be employed in sparsely gauged regions)? 2) The

model is claimed to be linear, but this is not the case (see specific comments below). and 3) The evaluations presented here are likely to be heavily influenced by the very high correlation (perfect in the case of one of the datasets) between the gauges used for adjustment and those used for validation. More specific remarks are given below.

**Specific remarks**

1. On p.3, line 24 the units of $\alpha$ are incorrect (should be mm h$^{-1}$ km$^{\beta}$ dB$^{-\beta}$).

2. On p.6, lines 10-12 it is mentioned that four links are selected. It's not clear to me what this selection was based on. I'm guessing that they were selected because these links were in (or close to) the catchment. Or were there more links in the area that were not selected. Can you provide a short statement in the paper about why these links were selected?

3. Section 2.3 seems redundant to me, and its contents can simply be put in Sections 2.1.1 and 2.1.2.

4. On p.7, lines 2-3 the authors claim that using the power law of Eq.(1) could result in overfitting. However, this power-law relation has been shown to be robust and relatively insensitive to variations in raindrop size distributions. So the parameters of this relation can be safely taken from literature without fitting them within a retrieval algorithm. The key to getting good rainfall estimates is to properly take effects of a variable baseline and wetting of antennas into account. So while I can certainly understand that the authors want to use an as simple equation as possible for the analyses presented in this paper, I think that the risk of overfitting should not be stated as a reason here.

5. On p.7, line 7 it is stated that $k$ is the specific attenuation after baseline separation. It would be good to specify here which method is used for determining and

separating this baseline.

6. On p.7, line 7, I suggest stating that you can use this simplification because $b$ is very close to 1 for the frequencies that are often used in CML networks.

7. On p.7, lines 20-21, as first glance I didn't think that it is necessary to state how the optimization is carried out because of the linearity of Eq.(2) and the fact that aggregation over time is a linear operation. Hence minimizing $L$ in Eq.(3) is a linear regression problem that has an analytical solution (even if you force the line to go through zero). However, I'm assuming that the authors are setting resulting rainfall estimates to zero if $k < k_w$ (which would yield $R < 0$ mm h$^{-1}$). This effectively means that although Eq.(2) is linear, the model that the authors are using is not. It should be expressed as

$$R = \begin{cases} \gamma\,(k - k_w) & \text{if} \quad k > k_w \\ 0 & \text{if} \quad k \le k_w. \end{cases}$$

I think that it should be clearly stated in the text that the model is effectively not linear. I also think that the implications of this nonlinearity should be discussed in the text. Furthermore, this means that the reason for using this linearized form that is stated by the authors is not valid (because they're using a nonlinear model). In fact, one could argue that Eq.(1) could be kept as a basis for the equation that is optimized, with a provision for correcting for wet antennas and baseline variations. Something like

$$R = \begin{cases} \alpha\,(k - k_w)^{\beta} & \text{if} \quad k > k_w \\ 0 & \text{if} \quad k \le k_w, \end{cases}$$

where $k_w$ includes wet antenna and baseline variation effects, and hence should then be the only parameter that is fitted (and $\alpha$ and $\beta$ taken from literature).

[Figure]

8. On p.7, line 31 a description is given on how the second parameter optimization run is carried out. It is stated that this run uses the parameter *distribution* of the first optimization run. However, I don't understand how the first run can yield a distribution of parameters. Or is it the distribution of parameters over all time steps in the entire dataset? In that case, the method cannot be used in a real-time setting.

9. On p.8, lines 7-12 it is stated that the effect of temporal aggregation is studied by comparing the gauge-adjusted CML rainfall product with the same gauges that were used to adjust the CML data. I expect the fact that the gauges are not dependent to have a large effect on the outcome of the analyses. Am I correct in assuming that this is only the case for the Dübendorf dataset, and that in Prague you're using the municipality gauge network as a reference? I think that the fact that the gauges in Switzerland are not independent should be discussed in the paper.

10. On p.8, line 24, a reference rainfall measurement is mentioned. It is not clear to me what this reference is. It this the average of the six (p.6 line 16) or four (Fig.1) rain gauges operated by the municipality for the Prague datset and the rain gauges and disdrometers for the Dübendorf dataset?

11. In Section 3.1 the authors discuss the reasons why parameter fitting for shorter intervals yield better results than for lnger intervals. I don't really agree with this discussion. What effectively happens when the length of the aggregation is increased is that the CML data receive more weight in determining the temporal evolution of the rainfall signal (relative to the gauges). Because either the same gauges (Dübendorf) or a gauge dataset that is well-correlated to the gauges that are used for the parameter fitting (Prague; see top-right panel of Fig.4) are used for verification, it is expected that the results are best if the weight of the gauges is largest (i.e., for the shortest accumulation intervals). So I don't think that you

can actually draw conclusions about which accumulation interval is best suited for this method based on these analyses.

12. On p.9, lines 20-22 the authors state that using daily rainfall accumulations to fit the model parameters would minimize the effect of diurnal fluctuations in baseline level. I think the converse is true: in order to minimize the effect of diurnal fluctuations, the model parameters should be fitted on a time scale that is significantly shorter than a day so that this variability is actually captured.

13. On p.13, Section 4.2 the authors discuss how the distribution of the $\gamma$ parameter changes with aggregation interval. This is then related to the fact that the proposed model includes the effect of wet antennas. However, this effect should be more related to the $k_w$ paramter of the model, and not so much to $\gamma$. Of course, the two model parameters can compensate, and this would result in wider distributions of $\gamma$, but this is a purely an effect of the fitting procedure.

14. On p.13, lines 17-18 the authors state that they've found a connection between the observed systematic errors and the degree of preservation of rainfall space-time structure through averaging. I don't really see this connection, and I think this should be better explained.

15. On p.14, line 9 the use of CML networks in sparsely gauged regions is mentioned. However, the method persented in this paper probably won't work in sparsely gauged reasons because rain gauges located close to the links are essential (see Figures 1, 4, and 5). So I think this statement needs to be altered.

16. On p.15, line 18 it is stated that CML networks can provide rainfall data on a (sub-)kilometer scale. However, I really don't think that this will be attainable with the method presented here. This is because of the fact that the CML data are adjusted to a (point) rain gauge somewhere in the vicinity, which will effectively

smooth out much of the variability captured by the individual links. So this statement should also be put into perspective.

17. In Figure 1, right panel, there seem to be white letters over the figure that are partly over the disdrometers.

18. In Figures 2, 5, and 6 the coefficient of determination ($R^2$) becomes negative. It would be good to give the definition of $R^2$ that was used in the paper in Section 2.6 (there are different versions of $R^2$, some of which cannot become negative).

19. In Figures 3 and 4 the slope of the regression line $y = ax$ (i.e., with fixed offset) is given. It should be noted here that the correlation between the two variables affects this slope. The slope will always be lower with a low correlation coefficient (you can try this by switching the $x$- and $y$-axes; see also the right-hand panels of Fig.4).
* * *

---

## Referee Comment (RC2) · S. Thorndahl (Referee) · 3 Oct 2016

The manuscript provides methods for adjusting rainfall estimates from commercial microwave links (CMLs) to rain gauges (RGs). It compares different temporal scales for adjustment and different layouts of gauge/CML systems. The work is novel and addresses very relevant issues in high resolution rainfall estimation in urban areas. It is well written and understandable and would fit well into the scope of HESS. Although not an expert in CMLs (but in radar rainfall estimation), I have some comments and suggestions which in my opinion could improve the manuscript.

1. It is unclear whether the paper aims for on-line (real-time) adjustment of CML's and thus real-time rainfall estimation or to estimate historical rainfall. Real-time adjustment would be associated with larger uncertainties. . . .

[Figure]

2. P4L31-P5L3: This is almost a conclusion of the paper. It does not belong in an introduction – but could be applied in the abstract.

3. In section 2, it should be argued why two different experimental sites are used. Could the same results not have been derived using only one site – or is there an objective to compare the two sites in terms of data, layout, etc.

4. During the paper it is also a bit confusing where averages of CMLs are used (in Prague) and when single CMLs are used. Please be clearer on this.

5. P6 bottom. It is unclear how you define an event. This is not necessarily an easy task operating with more than one rain sensor. Please clarify.

6. Section 2.6. You state that you adjust on different aggregation levels ranging from 5 min to 1 day, but compare on 1 minute values. Couldn't there be reason also to compare on larger aggregation levels than 1 min. It is well known that for small rain intensities rain gauges are not very accurate. E.g. one tip of 0.1 mm per minute in a tipping bucket rain gauge corresponds to 6 mm/h. An error of +/- 6 mm/h on gauge estimates over one minute for intensities larger than 6 mm/h, it therefore not unrealistic. For smaller intensities where the intensities are estimated using the time between two tips, the intensity at minute scale might be somewhat uncertain. In a paper (Thorndahl et al. 2004) we made radar-rain gauge adjustment over different temporal scales, but also compared the results over different scales. Maybe you could find some inspiration here.

7. With regards to estimating area rainfall (section 2.2 and 3.2) I guess results are still compared on the minute scale and adjustment is performend on larges temporal scales. I guess this will by associated with many random errors if there is rain in one gauge and not in another? Again I suggest to also comparing e.g. hourly estimates of rainfall.

8. Related to my comment no 4. I think it would be interesting to see a scatter plot of a

single CML vs a single RG and how R2 would depend on the range between CML and RG?

9. For the Dübendorf site it is unclear what you use the disdrometers for. Don't you use the RGs for adjustment/validation? Related to the problem above, disdrometers might be more accurate for small rain intensities?!

10. P9L18-19. A likely reason for the smaller scatter on the 1 day aggregation levels might be found in the fact that all of your events (except one) have duration shorter than 1 day. Thus, for some events same results for 12 and 24 h should be expected!

11. Figure 1. Please use lat/long or UTM rather than a local coordinate system.

References Thorndahl, S., Nielsen, J.E., Rasmussen, M.R., 2014. Bias adjustment and advection interpolation of long-term high resolution radar rainfall series. Journal of Hydrology 508, 214–226. doi:10.1016/j.jhydrol.2013.10.056

---

## Author Comment (AC1) · 1 Nov 2016

**General comments**

*Reviewer: This paper describes a method for incorporating accurate rain gauge measurements in commercial microwave link (CML) rainfall estimation through on-line parameter adjustment of the CML retrieval model. The idea of adjusting those model parameters that we know are most uncertain based on rain gauges is very appealing. This means that the accuracy of the gauges is used where it is most needed. The authors test their method on two different datasets, with different algorithm settings and different distances to the gauges used for adjustment. I think that the paper is interesting and certainly appropriate for HESS. I also have some issues that I think the authors should deal with before the paper is ready for publication. The most important*

*of these issues are: 1) How well does the presented method work when gauges are even further away from the links (i.e., how well can this method be employed in sparsely gauged regions)? 2) The model is claimed to be linear, but this is not the case (see specific comments below). 3) The evaluations presented here are likely to be heavily influenced by the very high correlation (perfect in the case of one of the datasets) between the gauges used for adjustment and those used for validation. More specific remarks are given below.*

Authors: It is very motivating for us that the reviewer acknowledges the scientific novelty of our study and its appropriateness for HESS. We also thank him for the very specific remarks, which will help us to minimize ambiguities in the presentation of the method and improve the clarity of the manuscript. Especially regarding the interpretation of the results. First, we address the general remarks. The detailed comments are then addressed in the "Specific remarks" section below each single comment.

1. **The distance of RGs to CMLs** represents an important limit for the use of our method. However, when RGs are far away this is limiting for any type of adjusting to ground observations, where "far" is conditional on the space-time correlation structure of rainfall. In our case, suitable distance of RGs to CMLs depends on the climatic conditions, type of rainfall (convective, frontal), the quality of CML data, and also application (requirements on time resolution). We discuss this, focusing especially on the limitations of our approach, in section 3.2 and 4.3. We discuss (p. 15, line 2–4) that already RG layouts covering areas in the range of 10–100 km$^2$ tend to underestimate rainfall peaks. We also suggest a potential remedy: where rain gauges are sparse, or even missing, short CMLs, which are often severely biased, could be adjusted to long CMLs, which more often behave according to wave propagation theory (p. 14, line 1–7). Although this is spec-ulative, because we did not test it in detail, it could be because, for long CMLs, there is relatively more water volume or drops in the propagation path than for

short CMLs. For short CMLs, the attenuation in the near field around the two end nodes, which is not well understood, is comparably larger. Unfortunately, although we believe that our dataset is truly unique, the RG information is not suitable for testing the method on more distant RGs. However, this does not invalidate the original goal of the presented manuscript, which is to show that adjusting CMLs by gauges is a feasible approach (even when using very straight-forward method) to improve space-time resolution of rainfall data, especially in urban areas. That said, we are, once more thankful for the reviewer's comments. We will take special care to better reflect the limits of the presented method (see specific remark 14).

2. **The general remark to the (non)linearity of the retrieval model** is addressed in detail under the specific remark 7. In the original manuscript we did not explicitly stated that the offset parameter $k_w$ is constrained to avoid model outputs with negative rainfall intensity. We also agree with reviewer that the model is not entirely linear, but piecewise-linear with two segments. We will clarify this in the manuscript.

3. **Regarding artefacts from high or perfect correlation between the RGs used for calibration and validation**, we are fully aware of the fact that the correlation between RGs constrains the efficiency of our approach. Despite of our effort to discuss this issue already in the initial version of the manuscript, some ambigu-ities clearly remain. The specific reviewers remarks were helpful to identify the corresponding paragraphs and improve the clarity of the text (please see remarks 9, 10, 11, and 19).

**Specific comments**

*1. On p. 3, line 24 the units of are incorrect (should be mm h$^{-1}$ km dB$^{-\beta}$).* Thank you, we will correct it.

*2. On p. 6, lines 10-12 it is mentioned that four links are selected. It's not clear to me what this selection was based on. I'm guessing that they were selected because these links were in (or close to) the catchment. Or were there more links in the area that were not selected. Can you provide a short statement in the paper about why these links were selected?*

Yes, we have selected links which correspond to the length scale of the catchment, i.e. to the reference rainfall. Thus, we have concentrated on CMLs which are shorter than two km (p. 7, lines 1-2, p. 13 lines 13–14). In our experience, this length is also the most relevant for applications in urban hydrology. Please also note that one CML was excluded from the analysis because connection was lost during the experiment. To clarify the selection we will add an additional figure in the supplementary material, which shows the map of the experimental catchment with the whole CML network of our collaborating partner, T-Mobile, as an overlay. We will refer to this material in section 2.1 Experimental sites (on p. 6, line 6). We will also add an information about which CMLs were affected by the data loss due to communication outages (on p. 6, line 12).

*3. Section 2.3 seems redundant to me, and its contents can simply be put in Sections 2.1.1 and 2.1.2.*

Agreed, we will put an information about experimental periods into sections 2.1.1 and 2.1.2 as suggested.

*4. On p.7, lines 2-3 the authors claim that using the power law of Eq.(1) could result in overfitting. However, this power-law relation has been shown to be robust and*

*relatively insensitive to variations in raindrop size distributions. So the parameters of this relation can be safely taken from literature without fitting them within a retrieval algorithm. The key to getting good rainfall estimates is to properly take effects of a variable baseline and wetting of antennas into account. So while I can certainly understand that the authors want to use an as simple equation as possible for the analyses presented in this paper, I think that the risk of overfitting should not be stated as a reason here.*

Thank you for this comment. In our revision, we will change the overfitting argument as suggested in comment 6, which addresses the same issue. In addition, we adjust section 4.1 to better discuss the potential and benefits of the suggested simplified relation.

*5. On p.7, line 7 it is stated that k is the specific attenuation after baseline separation. It would be good to specify here which method is used for determining and separating this baseline.*

Agreed, we will add this information. First, we make the common assumption that the baseline is constant during each wet period. Second, we classify the data into Dry and/wet periods. Classification is performed according to Schleiss et al. (2010) (using a moving window of length of 15 minutes). Third, we take the 10% quantile of the total path loss values in the preceding dry weather period as the best estimate.

*6. On p.7, line 7, I suggest stating that you can use this simplification because $b$ is very close to 1 for the frequencies that are often used in CML networks.*

We will add this "For frequencies between 20-40 GHz $\beta$ is relatively close to unity

according to ITU (2005) between 0.95 (20 GHz, vertical polarization) and 1.19 (40 GHz, horizontal polarization)."

*7. On p.7, lines 20-21, as first glance I didn't think that it is necessary to state how the optimization is carried out because of the linearity of Eq.(2) and the fact that aggregation over time is a linear operation. Hence minimizing L in Eq.(3) is a linear regression problem that has an analytical solution (even if you force the line to go through zero). However, I'm assuming that the authors are setting resulting rainfall estimates to zero if $k < k_w$ (which would yield $R < 0$ mm h$^{-1}$). This effectively means that although Eq.(2) is linear, the model that the authors are using is not. It should be expressed as*

$$R = \begin{cases} \gamma(k - k_w) & \text{if } k > k_w \\ 0 & \text{if } k \leq k_w \end{cases}$$

*I think that it should be clearly stated in the text that the model is effectively not linear. I also think that the implications of this nonlinearity should be discussed in the text. Furthermore, this means that the reason for using this linearized form that is stated by the authors is not valid (because they're using a nonlinear model). In fact, one could argue that Eq.(1) could be kept as a basis for the equation that is optimized, with a provision for correcting for wet antennas and baseline variations. Something like*

$$R = \begin{cases} \alpha(k - k_w)^\beta & \text{if } k > k_w \\ 0 & \text{if } k \leq k_w \end{cases}$$

*where $k_w$ includes wet antenna and baseline variation effects, and hence should then be the only parameter that is fitted (and and taken from literature).*

Thank you for this valuable remark. We used gradient-based optimization during the development of the technique, where we also tested other candidate models for which

analytical solutions were not available. To do this in an efficient manner, we used a single software implementation.

As suggested we will explicitly state in the revised manuscript that the tuning parameter $k_w$ is constrained, to avoid model to produce negative rainfall intensity. This means that $k_w$ cannot be higher than minimal specific attenuation $(k)$, i.e. for $k - k_w < 0$; $k_w = k$. We will, therefore, express the equation 2 as suggested by the reviewer. We also agree with reviewer that this means that model is not effectively linear in its whole domain, but piecewise linear. To avoid misunderstanding we will not call the model "linear" but "simplified". Finally, we will label the offset parameter $\Delta$ instead of $k_w$ to avoid misunderstandings and to emphasize that it is a general tuning parameter, which not only compensates for signal loss due to antenna wetting. We will also modify the description of the model (2) parameters (p. 7 lines 7-9) to: "where $\gamma$ [mm h$^{-1}$ km dB$^{-1}$] is an empirical parameter related to raindrop attenuation and other rainfall correlated signal losses, $k$ [dB km$^{-1}$] is a specific attenuation after baseline separation, and $\Delta$ [dB km$^{-1}$] is an offset parameter which corrects for wet antenna attenuation and possible bias introduced by inaccurate baseline identification. The piecewise linearity of the relation makes it possible to condition the model to rainfall and attenuation data which were aggregated over relatively long intervals (e.g., hours) and at the same time predict rainfall for attenuation data sampled at high frequencies."

It should be noted that, uncertainty related to attenuation from other effects than raindrop scattering and adsorption, i.e. "baseline variation effects" (including wet antenna effect) are most probably correlated with rainfall intensity and thus the $\Delta$ parameter cannot be uniquely optimized on its own. As stated in the section 4.1 (and also 4.2.): "The model (2) can be interpreted as a combination of linear forms of the attenuation-rainfall model (1) and WAA models". For details on why wet antenna attenuation cannot be, in our opinion, compensated by a single offset parameter please see response to comment 13.

*8. On p.7, line 31 a description is given on how the second parameter optimization run is carried out. It is stated that this run uses the parameter distribution of the first optimization run. However, I don't understand how the first run can yield a distribution of parameters. Or is it the distribution of parameters over all time steps in the entire dataset? In that case, the method cannot be used in a real-time Setting.*

This is not correct. It is correct that, in our analysis, we consider an "offline" setting, where we use the whole experimental period to estimate suitable parameter ranges. Thus, to use the method in near real-time setting the parameter ranges have to be estimated from past period. The continuous adjusting of model (2) does not "look into future", it uses rainfall intensity from the time step for which the adjustment is done and past rainfall intensities.

In summary, we will explicitly state in the corrected manuscript (in section 2.6) that we assess the method in the setting for historical rainfalls and we will also discuss the potential and limitation for real-time applications (for details please see our reply to comment 1 of the reviewer 2).

*9. On p.8, lines 7-12 it is stated that the effect of temporal aggregation is studied by comparing the gauge-adjusted CML rainfall product with the same gauges that were used to adjust the CML data. I expect the fact that the gauges are not dependent to have a large effect on the outcome of the analyses. Am I correct in assuming that this is only the case for the Dübendorf dataset, and that in Prague you're using the municipality gauge network as a reference? I think that the fact that the gauges in Switzerland are not independent should be discussed in the paper.*

Yes, this is correct. We intentionally investigated the effect of time aggregation by using the same RGs for conditioning and validation. This enables us to study the effect of

rainfall time averaging on the model's performance separately (without the influence of limited RG spatial representativeness). We investigate how performance degrades with increasing aggregation interval, e.g. due to averaging out of rainfall peaks or due to temporal evolution of the model parameters.

We state on p. 8, lines 8-11 of the original manuscript: "In this investigation, RGs used for CML adjustment are also reference RGs against which CMLs are evaluated. The only difference between rainfall used for adjusting and reference rainfall is the time resolution". As stated on p. 8, line 12 "The influence of RG layouts on CML adjusting is tested on Prague data only". Details of the analysis are further discussed under comment 11.

*10. On p.8, line 24, a reference rainfall measurement is mentioned. It is not clear to me what this reference is. It this the average of the six (p. 6 line 16) or four (Fig.1) rain gauges operated by the municipality for the Prague dataset and the rain gauges and disdrometers for the Dübendorf dataset?*

Thank you, we find this comment very helpful! It is important to distinguish unambiguously between rainfall used for adjusting and reference rainfall used as a "ground truth". We use the term "reference" for the rainfall to which we compare the best estimates from the adjusted CMLs. In the case of Prague, these are RGs located in the catchment, in the case of Dübendorf reference rainfall is taken as rainfall detected by the disdrometers along the CML path. In the first analysis where the effect of rainfall aggregation is investigated, we use the same RGs (resp. disdrometers) for adjusting and the same RGs as reference. We only used them at different temporal scales. In the second analysis (on Prague dataset only), where influence of RG spacing is tested, three different spatial layouts are used for adjusting, however we still use the reference RGs in the catchment for performance evaluation of the estimates from adjusted CMLs, i.e. we use same reference rainfall as in the first analysis.
We will clarify this issue on p. 6, lines 14-18 and lines 20-24. In addition, we will correct the typo on p. 6 lines 15-17, where we mistakenly referred to Fig. 1, left instead of middle and vice versa. Furthermore, we will add reference to the Fig. 1, on p. 8 line 9 and 12 (performance assessment section) to clarify which rainfall is used as the reference.

*11. In Section 3.1 the authors discuss the reasons why parameter fitting for shorter intervals yield better results than for longer intervals. I don't really agree with this discussion. What effectively happens when the length of the aggregation is increased is that the CML data receive more weight in determining the temporal evolution of the rainfall signal (relative to the gauges). Because either the same gauges (Dübendorf) or a gauge dataset that is well-correlated to the gauges that are used for the parameter fitting (Prague; see top-right panel of Fig. 4) are used for verification, it is expected that the results are best if the weight of the gauges is largest (i.e., for the shortest accumulation intervals). So I don't think that you can actually draw conclusions about which accumulation interval is best suited for this method based on these analyses.*

In section 3.1 we do not investigate optimal temporal aggregation intervals as stated already in the performance assessment section (see comment 9). We only study, how model performance worsens with increasing aggregation interval and we try to relate it to the autocorrelation structure of rainfall. We are very much aware of the fact that shorter aggregation intervals give more weights to the gauges and therefore, the best performance has to be achieved by short intervals when same gauges are used for adjusting and evaluation. The optimal aggregation interval is investigated subsequently in section 3.2 where aggregation is used to improve spatial representativeness of RGs far away from the catchment, resp. CMLs.

In our view, this is a misunderstanding, which partly arises from the wrong cross-references to Figure 1, which will be fixed (please see also our previous response).
In addition, we will add into performance assessment section a short paragraph explaining the goal of the time averaging analysis, which is: i) to indicate limits of the proposed method for disaggregating cumulative rainfall data (15 min, hourly, daily), which are at many places available, unlike minute data. ii) to ease interpretation of CML adjustment by different RG layouts (section 3.2), where temporal aggregation is used to improve the spatial representativeness of the RGs.

*12. On p.9, lines 20-22 the authors state that using daily rainfall accumulations to fit the model parameters would minimize the effect of diurnal fluctuations in baseline level. I think the converse is true: in order to minimize the effect of diurnal fluctuations, the model parameters should be fitted on a time scale that is significantly shorter than a day so that this variability is actually captured.*

We agree and we will remove this statement.

*13. On p.13, Section 4.2 the authors discuss how the distribution of the $\gamma$ parameter changes with aggregation interval. This is then related to the fact that the proposed model includes the effect of wet antennas. However, this effect should be more related to the $k_w$ parameter of the model, and not so much to $\gamma$. Of course, the two model parameters can compensate, and this would result in wider distributions of $\gamma$, but this is a purely an effect of the fitting procedure.*

We have a different opinion on this particular issue and presume this is rather a misunderstanding caused by the unfortunate naming the offset parameter "$k_w$" (in the future $\Delta$) of model (2) (see reply to comment 7) and its imprecise description on p. 7, lines 7-9. This might create the false impression that only the offset parameter is responsible for wet antenna attenuation (WAA) correction. It is important to note that the simpli-
fied model (2) should be interpreted as "a combination of WAA model and simplified standard power-law model" (p. 13 lines 9-10). Although Overeem et al. 2011 suggest that, for their 15 minute CML data, WAA can be satisfactorily modelled as a constant, the other authors suggest more complex models. Given our theoretical understanding, these should generally depend on rainfall intensity (e.g. Kharadly et al. 2001, Leijnse et al. 2008). Indeed, we found that WAA react very dynamically on changes in rainfall intensity. Spraying the radomes of some radios in Prague showed a substantial dynamic response. Immediately after spraying, attenuation increased by about 5 dB, decreased to 2.5 dB after 1 minute, and was almost not observable any more after 3-4 minutes (Fig. 1, this response).

If WAA depends on rainfall intensity, the linear approximation of any WAA model which reflects this dependence then should be affected by compensation of the offset parameter by the slope parameter. This also would explain (p. 13, lines 8-10) the discrepancy between $\gamma$ parameters of model (2) and $\alpha$ parameter of model (1) suggested by ITU (2005). Such discrepancy was already reported by Fenicia et. al (2012) "who estimated for their 23 GHz CML values of $\alpha$ substantially lower than values suggested by ITU (2005)" (p. 13, lines 10-11).

To clarify the nature of the simplified model (Eq. 2) and avoid misunderstanding, we suggest to label $k_w$ as $\Delta$ instead. And also change the description of the parameters when first introducing model(2) (p. 7, lines 7 - 9). Third, we will better explain that the simplified model combines linear approximations of both the rainfall retrieval model (1) and wet antenna model in section 2.4.

*14. On p.13, lines 17-18 the authors state that they've found a connection between the observed systematic errors and the degree of preservation of rainfall space-time structure through averaging. I don't really see this connection, and I think this should be better explained.*

We explained this connection in section 3.1 p. 9 lines 11-16 of the original manuscript, and we showed in the appendix A (and figure A1 in the manuscript) how increasing the aggregation interval smoothes out rainfall peaks and smoothes out the differences between low and high intense rainfall periods. In our opinion, this smoothing of rainfall peaks most likely explains why the identification of model (2) parameters worsens with increasing aggregation intervals. Although we did not formally describe the relation between preservation of correlation patterns in aggregated rainfall and model parameter identification, we sufficiently demonstrate that this relation exist and thus we can explicitly state on p. 13 lines 17-18 that our results suggest that the underestimation of peak intensities is influenced by the preservation of autocorrelation in the aggregated rainfall (Fig. A1, in the manuscript).

*15. On p.14, line 9 the use of CML networks in sparsely gauged regions is mentioned. However, the method presented in this paper probably won't work in sparsely gauged reasons because rain gauges located close to the links are essential (see Figures 1, 4, and 5). So I think this statement needs to be altered.*

Thank you for the comment. It is also discussed at p. 14 lines 1-7 and p. 15 lines 10-11), however, we agree that the presented analyses rather is a proof-of concept than enables us to generalize and extrapolate to different conditions, e.g. RG layout, topology, climate, weather, ...). In particular extrapolation to sparsely gauged regions has to be performed with great care. We will, therefore, modify the first sentence to: "Commercial microwave links (CMLs) can improve resolution of existing rain gauge and radar networks, especially in populated areas where there are often very dense."

*16. On p.15, line 18 it is stated that CML networks can provide rainfall data on a (sub-)kilometer scale. However, I really don't think that this will be attainable with the method presented here. This is because of the fact that the CML data are adjusted to*

*a (point) rain gauge somewhere in the vicinity, which will effectively smooth out much of the variability captured by the individual links. So this statement should also be put into perspective.*

Thank you for this comment, we have considered it carefully. Nevertheless, to our opinion combined use of RGs and CMLs can provide "insight into rainfall space-time structure at (sub)minute and (sub)kilometer scale" (p. 15 lines 18-19). We have demonstrated in presented analyses that even CMLs with sub-kilometer path lengths are, after adjusting, capable to provide accurate rainfall estimates outperforming RGs used for adjusting. In our investigation we poll CMLs with approx. 10 s resolution and it is technically possible to poll CMLs at (sub)second resolution (e.g. Chwala et al. 2016), although this might also be influenced by the firmware and hardware of the radio. Moreover, the CML networks especially in city centres can be very dense, in the Prague (CZ) city centre it is up to 50 CMLs per km$^2$. We, therefore, think it is appropriate to conclude that CMLs can provide "insight into rainfall space-time structure at (sub)kilometer and (sub)minute scale", although we are aware of the fact that adjusting can lead to averaging of rainfall peaks etc. This is, however, also happening when adjusting weather radar rainfall data and they are commonly used to estimates rainfall space-time structure at subkilometer scale.

*17. In Figure 1, right panel, there seem to be white letters over the figure that are partly over the disdrometers.*

Thank you, we will correct this.

*18. In Figures 2, 5, and 6 the coefficient of determination ($R^2$) becomes negative. It would be good to give the definition of $R^2$ that was used in the paper in Section 2.6*

*(there are different versions of $R^2$, some of which cannot become negative).*

Thank you. We used pseudo $R^2$ as defined by Efron (1978), i.e. it is defined identically as the Nash-Sutcliffe efficiency ($NSE$), a popular measure in (urban)hydrology. We will change the label in the whole manuscript to $NSE$ to avoid misunderstanding.

*19. In Figures 3 and 4 the slope of the regression line y = ax (i.e., with fixed offset) is given. It should be noted here that the correlation between the two variables affects this slope. The slope will always be lower with a low correlation coefficient (you can try this by switching the x- and y-axes; see also the right-hand panels of Fig.4).*

This is another valuable remark. We suggest to add correlation coefficients into the legends of scatterplots in both figures (see Fig. 2 in this response).

The correlation coefficients on Figure 2 (in this response) shows, that even CMLs adjusted to remote RGs correlate very well with reference rainfall. The slope of CML-reference rainfall regression line is therefore rather influenced by systematic underestimation of rainfall peaks. In contrast to that, the correlation between RG layouts which cover larger areas and reference rainfalls are much lower (at 1 min resolution), which indeed affects the slope of regression lines. However, aggregating the RG intensities over longer intervals increases the correlation. Consequently, a longer aggregation interval improves the performance of the adjusting algorithm compared to shorter aggregation (see Fig. 5 in the manuscript: in the case of layout B2 with relatively distant RGs - the $NSE$ of 1h ranges between 0.50-0.91 with median 0.78, whereas 5 min only achieves $NSE$ between 0.2-0.86 with median 0.75). The areal averaging leads, however, to the smoothing out of rainfall peaks which in turn systematically affects peak rainfalls estimated from adjusted CMLs.

References:

Chwala, C., Keis, F. and Kunstmann, H.: Real-time data acquisition of commercial microwave link networks for hydrometeorological applications, Atmos Meas Tech, 9(3), 991–999, doi:10.5194/amt-9-991-2016, 2016.

Efron, B.: Regression and Anova with Zero-One Data - Measures of Residual Variation, J. Am. Stat. Assoc., 73(361), 113–121, doi:10.2307/2286531, 1978.

Fenicia, F., Pfister, L., Kavetski, D., Matgen, P., Iffly, J.-F., Hoffmann, L. and Uijlenhoet, R.: Microwave links for rainfall estimation in an urban environment: Insights from an experimental setup in Luxembourg-City, J. Hydrol., 464–465, 69–78, doi:10.1016/j.jhydrol.2012.06.047, 2012.

ITU: ITU-R P.838-3, [online] Available from: http://www.itu.int/dms_pubrec/itu-r/rec/p/R-REC-P.838-3-200503-I!!PDF-E.pdf, 2005.

Kharadly, M. M. Z. and Ross, R.: Effect of wet antenna attenuation on propagation data statistics, Antennas Propag. IEEE Trans. On, 49(8), 1183–1191, 2001.

Leijnse, H., Uijlenhoet, R. and Stricker, J. N. M.: Microwave link rainfall estimation: Effects of link length and frequency, temporal sampling, power resolution, and wet antenna attenuation, Adv. Water Resour., 31(11), 1481–1493, 2008.

Overeem, A., Leijnse, H. and Uijlenhoet, R.: Measuring urban rainfall using microwave links from commercial cellular communication networks, Water Resour. Res., 47, 16

PP., doi:201110.1029/2010WR010350, 2011.

Schleiss, M. and Berne, A.: Identification of Dry and Rainy Periods Using Telecommunication Microwave Links, IEEE Geosci. Remote Sens. Lett., 7(3), 611–615, doi:10.1109/LGRS.2010.2043052, 2010.

[Figure]

**Fig. 1.** Wet antenna attenuation of about 5 dB for a 38 GHz CML almost disappears within 3-4 minutes after spraying the antenna radome during dry weather.

[Figure]

**Fig. 2.** Revision of the figure 4 in the original manuscript.

---

## Author Comment (AC2) · 1 Nov 2016

**General comments**

Reviewer: The manuscript provides methods for adjusting rainfall estimates from commercial microwave links (CMLs) to rain gauges (RGs). It compares different temporal scales for adjustment and different layouts of gauge/CML systems. The work is novel and addresses very relevant issues in high resolution rainfall estimation in urban areas. It is well written and understandable and would fit well into the scope of HESS. Although not an expert in CMLs (but in radar rainfall estimation), I have some comments and suggestions which in my opinion could improve the manuscript.

Authors: First, we would like to thank reviewer for all the remarks and recommendations

how to complete the manuscript and improve its clarity. Clearly, an expert on weather radars experienced in adjusting to rain gauges can give substantial advice.

**Specific comments**

1. It is unclear whether the paper aims for on-line (real-time) adjustment of CML's and thus real-time rainfall estimation or to estimate historical rainfall. Real-time adjustment would be associated with larger uncertainties.

In our analysis we assess the method in the setting for historical rainfalls. However, the method does not "look into future" when continuously adjusting model (2), but uses rainfall intensity from the time point for which the adjustment is done and then from several time points in the past (P7L14-24). Thus, method can be used with additional tuning in near real-time setting.

To have a better evidence base, we performed additional analyses where CML rainfall retrieval model is continuously adjusted based only on past data. The results of these analyses confirmed that use of the method in real-time setting leads to only slightly worse CML performance in comparison to the original analysis. We therefore suggest to explicitly state in the section 2.6 that we assess the method in the setting for historical rainfalls and discuss the use of the method for real-time setting in the Discussion section.

2. P4L31-P5L3: This is almost a conclusion of the paper. It does not belong in an introduction – but could be applied in the abstract.

Thank you for this comment. In our view, this paragraph improves the understandability and clarity of a manuscript to i) have a very specific message and ii) convey the message to the reader. This can include explicitly stating the novelty of the work, but also concrete results. Then, a reader is not left in the dark what to expect and will not have major surprises - which are always confusing - during reading. As the HESSD
abstract is too short, this info can go into the introduction. In our view, the redundancy of information-pieces (twice mentioned in the abstract and the intro) is a small price to pay for the increased clarity.

3. In section 2, it should be argued why two different experimental sites are used. Could the same results not have been derived using only one site – or is there an objective to compare the two sites in terms of data, layout, etc.

Thank you for this comment. The reasons behind using data from two sites are distributed over the two sections 2.1.1 and 2.1.2 (differences in operational mode P5L16-17, P5L22-24, P6L1-2, or P6L12-13) and they include different reference rainfall data and power-control settings of CMLs on each of the sites. We agree that this is sub-optimal. We suggest to briefly mention our main reasons in the introductory paragraph of the section 2.1: "We deliberately analyze datasets from two different experimental sites. This enables us to test the feasibility of the proposed approach on CMLs operated with and without automatic power control. Moreover, the dataset from Dübenbdorf contains detailed reference rainfall measurements along a CML path, which provide very good basis for investigating specifics of a rainfall from a single CML. In contrast, the areal rainfall observations from Prague are more appropriate to analyze rainfall retrieval from multiple CMLs, which is also more relevant to evaluate the proposed adjusting method for common urban hydrological applications."

4. During the paper it is also a bit confusing where averages of CMLs are used (in Prague) and when single CMLs are used. Please be clearer on this.

Thanks, we now see that this information is indeed missing in the method section (2.6 performance assessment), but we present it in the Result section (P93-4 and P9L31-P10L3) instead. We will therefore improve section 2.6., P9, line 19.
5. P6 bottom. It is unclear how you define an event. This is not necessarily an easy task operating with more than one rain sensor. Please clarify.

The events at both experimental sites are first defined from each of the sensors and then the event periods are merged by simply increasing the duration to include the very first and the very last observation of a sensor. In the case of Dübendorf the events were defined based on disdrometer classification. In the case of Prague, events were defined from reference RG measurements. The beginning of an event is assumed to be 15 minutes before the first tip of RG and end of event 15 minutes after the last tip. In addition, the beginnings and ends of the events in Prague area were rounded (down resp. up) to full hours to ease the analysis with aggregated rainfall intensities. At both sites two rainy periods separated by shorter interval than 30 minutes are assumed to be the same event.

We will add an information about event definition at the end of the paragraph in section 2.3. Note, however, that adjusting is performed over whole experimental period and thus it is independent of event definition. The event definition therefore influences the performance evaluation, i.e. by the (non-)selection of events.

6. Section 2.6. You state that you adjust on different aggregation levels ranging from 5 min to 1 day, but compare on 1 minute values. Couldn't there be reason also to compare on larger aggregation levels than 1 min. It is well known that for small rain intensities rain gauges are not very accurate. E.g. one tip of 0.1 mm per minute in a tipping bucket rain gauge corresponds to 6 mm/h. An error of +/- 6 mm/h on gauge estimates over one minute for intensities larger than 6 mm/h, it therefore not unrealistic. For smaller intensities where the intensities are estimated using the time between two tips, the intensity at minute scale might be somewhat uncertain. In a paper (Thorndahl et al. 2014) we made radar-rain gauge adjustment over different temporal scales, but also compared the results over different scales. Maybe you could find some inspiration here.
Thank you for this suggestion. We used one minute reference data because this is the temporal resolution required for rainfall-runoff modeling at the scale of small urban catchments and our long-term intention is to provide rainfall data which could be used for this purpose. In the case of Dubendorf site, disdrometers are very well suited for providing rainfall data at 1 min resolution. The sampling error of tipping bucket RGs in Prague is partly reduced by calculating areal rainfall from six RGs relatively close to each other. Nevertheless, we agree that this sampling error may influence  $R^2$  values. We have therefore compared rainfall estimates from CML data also at other temporal scales (Fig 1, this response).

We find only small changes in  $R^2$  values when comparing CML rainfall to reference rainfall at larger temporal scales. This indicates that the analysis even at 1 min scale is not substantially influenced by random errors. Interestingly, we see a slight increase in  $R^2$  values for larger temporal scales of reference rainfall, although aggregation should reduce RG sampling errors. In our view, giving a larger weight to the RG data in the adjusting procedure increases the  $R^2$ , because the temporal scale of reference rainfall gets closer to the aggregation interval used for CML adjusting (for details see comment no. 11 of the reviewer 1). The results presented here do not, however, change our conclusions drawn in the original manuscript where we only presented the performance for 1 minute data.

7. With regards to estimating area rainfall (section 2.2 and 3.2) I guess results are still compared on the minute scale and adjustment is performed on larges temporal scales. I guess this will be associated with many random errors if there is rain in one gauge and not in another? Again I suggest to also comparing e.g. hourly estimates of Rainfall.

Yes, the discrepancy between rainfall measured by those RG layouts which were used for adjusting and those used for validation purposes (reference rainfall), indeed influences the performance of the adjusting procedure. Here, we reduce these errors by aggregating the RG data used for adjusting to longer intervals (up to 1 h). The
performance of the procedure is then evaluated by comparing adjusted CMLs to the reference rainfall (i.e. RGs in the catchment). Thus, all rainfall observation errors which stem from RG layouts (including instrumental errors, sampling errors and limited spatial representativeness of point RG measurements) are implicitly included in the evaluation. The comparison at larger time scales would indeed reduce the sampling error in reference rainfall. However, as discussed in the reply above, their influence on the performance is small. Moreover, our adjusting procedure is only relevant where the temporal scale of reference rainfall (resp. adjusted rainfall) is finer than the aggregation interval used for CML adjusting. As we identified in our analysis that optimal aggregation intervals for the evaluated RG layouts are relatively short (15 min for layouts A1 resp. B1 and 1 h for layout B2), the comparison to e.g. hourly estimates is not useful.

8. Related to my comment no 4. I think it would be interesting to see a scatter plot of a single CML vs a single RG and how  $R^2$  would depend on the range between CML and RG?

Thank you for this suggestion. Unfortunately, although this might be an interesting analysis our experimental layout is not suitable for that. Each CML included in the analysis (see Fig. 1 in the manuscript) has different features (lengths, frequencies, polarizations) which considerably influences its performance in terms of rainfall estimation. The differences between single CMLs and a corresponding RG would be dominated by these differences. In our experience, the discrepancy between path integrated and point measured rainfall usually dominates the discrepancy due to different locations of the CML and the RG.

9. For the Dübendorf site it is unclear what you use the disdrometers for. Don't you use the RGs for adjustment/validation? Related to the problem above, disdrometers might be more accurate for small rain intensities?!
Thank you, we will modify the sentence at P5L24-26 to: "In addition to the five disdrometers used in our analyses to retrieve reference rainfall, three tipping bucket RGs measure rainfall intensities, which makes it possible to validate the disdrometer data."

10. P9L18-19. A likely reason for the smaller scatter on the 1 day aggregation levels might be found in the fact that all of your events (except one) have duration shorter than 1 day. Thus, for some events same results for 12 and 24 h should be expected!

In our study we adjusted each CML over the whole experimental period, although it is evaluated only on events listed in the table 1. Thus for longer aggregation intervals also other events (with heights lower than 5 mm) influence the adjusting. This is also one of the reasons why CML adjusted with 12 h aggregation interval have different scatter than adjusted to 1 d aggregation interval. We will write more clearly that CMLs are adjusted over the whole experimental periods in the section 2.6, performance assessment.

**11. Figure 1. Please use lat/long or UTM rather than a local coordinate system.**

Thank you for the suggestion. We will change it as shown in the Fig. 2 in this response.

**References:**

Thorndahl, S., Nielsen, J.E., Rasmussen, M.R., 2014. Bias adjustment and advection interpolation of long-term high resolution radar rainfall series. Journal of Hydrology 508, 214–226. doi:10.1016/j.jhydrol.2013.10.056
**Fig. 1.** Comparison of R2 for different temporal aggregations of reference rainfall. Mean of four CMLs (see figure 2 of the original manuscript) adjusted to rainfall having different aggregation intervals

---

## Author Response (AR1)

**Reply to Reviewer 1: H. Leijnse**

**General comments**

Reviewer: This paper describes a method for incorporating accurate rain gauge measurements in commercial microwave link (CML) rainfall estimation through on-line parameter adjustment of the CML retrieval model. The idea of adjusting those model parameters that we know are most uncertain based on rain gauges is very appealing. This means that the accuracy of the gauges is used where it is most needed. The authors test their method on two different datasets, with different algorithm settings and different distances to the gauges used for adjustment. I think that the paper is interesting and certainly appropriate for HESS. I also have some issues that I think the authors should deal with before the paper is ready for publication. The most important of these issues are: 1) How well does the presented method work when gauges are even further away from the links (i.e., how well can this method be employed in sparsely gauged regions)? 2) The model is claimed to be linear, but this is not the case (see specific comments below). 3) The evaluations presented here are likely to be heavily influenced by the very high correlation (perfect in the case of one of the datasets) between the gauges used for adjustment and those used for validation. More specific remarks are given below.

Authors: It is very motivating for us that the reviewer acknowledges the scientific novelty of our study and its appropriateness for HESS. We also thank him for the very specific remarks, which will help us to minimize ambiguities in the presentation of the method and improve the clarity of the manuscript. Especially regarding the interpretation of the results. First, we address the general remarks. The detailed comments are then addressed in the "Specific remarks" section below each single comment. The reviewer comments refer to the line numbers and section numbers in the original manuscript, however our responses refer already to the revised manuscript. As we slightly restructured the structure of the manuscript, section numbers might differ.

1. **The distance of RGs to CMLs** represents an important limit for the use of our method. However, when RGs are far away this is limiting for any type of adjusting to ground observations, where "far" is conditional on the space-time correlation structure of rainfall. In our case, suitable distance of RGs to CMLs depends on the climatic conditions, type of rainfall (convective, frontal), the quality of CML data, and also application (requirements on time resolution). We discuss this, focusing especially on the limitations of our approach, in section 3.2 and 4.3. We discuss (p. 15, line 2–4) that already RG layouts covering areas in the range of 10–100 km$^2$ tend to underestimate rainfall peaks. We also suggest a potential remedy: where rain gauges are sparse, or even missing, short CMLs, which are often severely biased, could be adjusted to long CMLs, which more often behave according to wave propagation theory (p. 14, line 1–7). Although this is speculative, because we did not test it in detail, it could be because, for long CMLs, there is relatively more water volume or drops in the propagation path than for short CMLs. For short CMLs, the attenuation in the near field around the two end nodes, which is not well understood, is comparably larger. Unfortunately, although we believe that our dataset is truly unique, the RG information is not suitable for testing the method on more distant RGs. However, this does not invalidate the original goal of the presented manuscript, which is to show that adjusting CMLs by gauges is a feasible approach (even when using very straight-forward method) to improve space-time resolution of

rainfall data, especially in urban areas. That said, we are, once more thankful for the reviewer's comments. We will take special care to better reflect the limits of the presented method (see specific remark 14).

2. **The general remark to the (non)linearity of the retrieval model** is addressed in detail under the specific remark 7. In the original manuscript we did not explicitly stated that the offset parameter $k_w$ is constrained to avoid model outputs with negative rainfall intensity. We also agree with reviewer that the model is not entirely linear, but piecewise-linear with two segments. We will clarify this in the manuscript.

3. **Regarding artefacts from high or perfect correlation between the RGs used for calibration and validation**, we are fully aware of the fact that the correlation between RGs constrains the efficiency of our approach. Despite of our effort to discuss this issue already in the initial version of the manuscript, some ambigu-ities clearly remain. The specific reviewers remarks were helpful to identify the corresponding paragraphs and improve the clarity of the text (please see remarks 9, 10, 11, and 19).

**Specific comments**

**Q1:** On p. 3, line 24 the units of are incorrect (should be mm h$^{-1}$ km dB$^{-\beta}$).
**A1:** Thank you, corrected.

**Q2:** On p. 6, lines 10-12 it is mentioned that four links are selected. It's not clear to me what this selection was based on. I'm guessing that they were selected because these links were in (or close to) the catchment. Or were there more links in the area that were not selected. Can you provide a short statement in the paper about why these links were selected?
**A2:** Yes, we have selected links which correspond to the length scale of the catchment, i.e. to the reference rainfall. Thus, we have concentrated on CMLs which are shorter than two km (p. 7, lines 1-2, p. 13 lines 13–14 in the original manuscript). In our experience, this length is also the most relevant for applications in urban hydrology. Please also note that one CML was excluded from the analysis because connection was lost during the experiment. To clarify the selection we have added an additional figure in the supplementary material, which shows the map of the experimental catchment with the whole CML network of our collaborating partner, T-Mobile, as an overlay. We refer to this material in section 2.1 Experimental sites. We also added into this subsection an information about excluded CMLs due to communication outages.

**Q3:** Section 2.3 seems redundant to me, and its contents can simply be put in Sections 2.1.1 and 2.1.2.
**A3:** We have removed section 2.3. and put an information about experimental periods into sections 2.1.1 and 2.1.2 and partly also into performance assessment section now referred as 2.5.

**Q4:** On p.7, lines 2-3 the authors claim that using the power law of Eq.(1) could result in overfitting. However, this power-law relation has been shown to be robust and relatively insensitive to variations in raindrop size distributions. So the parameters of this relation can be safely taken from literature without fitting them within a retrieval algorithm. The key to getting good rainfall estimates is to properly take effects of a variable baseline and wetting of antennas into account. So while I can certainly understand that the authors want to use an as simple equation as possible for the analyses presented in this paper, I think that the risk of overfitting should not be stated as a reason here.

**A4:** Thank you for this comment. We changed the overfitting argument as suggested in comment 6, which addresses the same issue. We also changed this argument in the introduction. In addition, we slightly adjusted section 4.1 where simplified model is discussed.

**Q5:** On p.7, line 7 it is stated that $k$ is the specific attenuation after baseline separation. It would be good to specify here which method is used for determining and separating this baseline.

**A5:** Agreed, we added at the end of the section 2.3 this information: "The baseline for specific attenuation $k$ is assumed to be constant during each wet periods. First, we classify the data into dry and wet periods. Classification is performed according to Schleiss et al. (2010) (using a moving window of length of 15 minutes). Second, we take the 10% quantile of the total path loss values in the preceding dry weather period as the best estimate."

**Q6:** On p.7, line 7, I suggest stating that you can use this simplification because b is very close to 1 for the frequencies that are often used in CML networks.

**A6:** We added this information into introducing paragraph of the section 2.4 "For frequencies between 20-40 GHz $\beta$ is relatively close to unity according to ITU (2005) between 0.95 (20 GHz, vertical polarization) and 1.19 (40 GHz, horizontal polarization)."

**Q7:** On p.7, lines 20-21, as first glance I didn't think that it is necessary to state how the optimization is carried out because of the linearity of Eq.(2) and the fact that aggrega-tion over time is a linear operation. Hence minimizing L in Eq.(3) is a linear regression problem that has an analytical solution (even if you force the line to go through zero). However, I'm assuming that the authors are setting resulting rainfall estimates to zero if $k < k_w$ (which would yield R < 0 mm h$^{-1}$). This effectively means that although Eq.(2) is linear, the model that the authors are using is not. It should be expressed as

$$R = \begin{cases} \alpha(k - k_w)^\beta & \text{if } k > k_w \\ 0 & \text{if } k \leq k_w \end{cases}$$

I think that it should be clearly stated in the text that the model is effectively not linear. I also think that the implications of this nonlinearity should be discussed in the text. Furthermore, this means that the reason for using this linearized form that is stated by the authors is not valid (because they're using a nonlinear model). In fact, one could argue that Eq.(1) could be kept as a basis for the equation that is optimized, with a provision for correcting for wet antennas and baseline variations. Something like

$$R = \begin{cases} \gamma(k - k_w) & \text{if } k > k_w \\ 0 & \text{if } k \leq k_w \end{cases}$$

where $k_w$ includes wet antenna and baseline variation effects, and hence should then be the only parameter that is fitted (and and taken from literature).

**A7:** Thank you for this valuable remark. We used gradient-based optimization during the development of the technique, where we also tested other candidate models for which analytical solutions were not available. To do this in an efficient manner, we used a single software implementation.

As suggested we explicitly stated in the revised manuscript that the tuning parameter $k_w$ (now referred as Δ) is constrained, to avoid model to produce negative rainfall intensity and expressed the equation 2 as suggested by the reviewer. We also agree with reviewer that this means that model is not effectively linear in its whole domain, but piecewise linear. To avoid misunderstanding we do not call the model "linear" anymore, but "simplified". Finally, we labeled the offset parameter Δ instead of $k_w$ to avoid misunderstandings and to emphasize that it is a general tuning parameter, which not only compensates for signal loss due to antenna wetting. We will also modify the description of the model (2) parameters (p. 7 lines 14-21) to: "where γ [mm h$^{-1}$ km dB$^{-1}$] is an empirical parameter related to raindrop attenuation and other rainfall correlated signal losses, $k$ [dB km$^{-1}$] is a specific attenuation after baseline separation, and Δ [dB km$^{-1}$] is an offset parameter which corrects for wet antenna attenuation and possible bias introduced by inaccurate baseline identification. The parameter is constrained, to avoid model to produce negative rainfall intensity. The piecewise linearity of the relation makes it possible to condition the model to rainfall and attenuation data which were aggregated over relatively long intervals (e.g., hours) and at the same time predict rainfall for attenuation data sampled at high frequencies."

It should be noted that, uncertainty related to attenuation from other effects than raindrop scattering and adsorption, i.e. "baseline variation effects" (including wet antenna effect) are most probably correlated with rainfall intensity and thus the parameter cannot be uniquely optimized on its own. As stated in the section 4.1 (and also 4.2.): "The model (2) can be interpreted as a combination of linear forms of the attenuation-rainfall model (1) and WAA models". For details on why wet antenna attenuation cannot be, in our opinion, compensated by a single offset parameter please see response to comment 13.

**Q8:** On p.7, line 31 a description is given on how the second parameter optimization run is carried out. It is stated that this run uses the parameter distribution of the first optimization run. However, I don't understand how the first run can yield a distribution of parameters. Or is it the distribution of parameters over all time steps in the entire dataset? In that case, the method cannot be used in a real-time Setting.

**A8:** This is not correct. It is correct that, in our analysis, we consider an "offline" setting, where we use the whole experimental period to estimate suitable parameter ranges. Thus, to use the method in near real-time setting the parameter ranges have to be estimated from past period. The continuous adjusting of model (2) does not "look into future", it uses rainfall intensity from the time step for which the adjustment is done and past rainfall intensities.

In summary, we explicitly stated in the corrected manuscript (in section 2.5) that we assessed the method in the setting for historical rainfalls and we also discuss the potential and limitation for real-time applications in the new section 4.3 (for more details on real-time capability of our algorithm please see our reply to comment 1 of the reviewer 2).

**Q9:** On p.8, lines 7-12 it is stated that the effect of temporal aggregation is studied by comparing the gauge-adjusted CML rainfall product with the same gauges that were used to adjust the CML data. I expect the fact that

the gauges are not dependent to have a large effect on the outcome of the analyses. Am I correct in assuming that this is only the case for the Dübendorf dataset, and that in Prague you're using the municipality gauge network as a reference? I think that the fact that the gauges in Switzerland are not independent should be discussed in the paper.

**A9:** Yes, this is correct. We intentionally investigated the effect of time aggregation by using the same RGs for conditioning and validation. This enables us to study the effect of rainfall time averaging on the model's performance separately (without the influence of limited RG spatial representativeness). We investigate how performance degrades with increasing aggregation interval, e.g. due to averaging out of rainfall peaks or due to temporal evolution of the model parameters.

To make this clear, we have added into the performance assessment section this information: "… we explore whether the proposed adjusting method can be used to disaggregate cumulative rainfall data, such as hourly or daily values, to one-minute data."

Details of the analysis are further discussed under comment 11.

**Q10:** On p.8, line 24, a reference rainfall measurement is mentioned. It is not clear to me what this reference is. It this the average of the six (p. 6 line 16) or four (Fig.1) rain gauges operated by the municipality for the Prague dataset and the rain gauges and disdrometers for the Dübendorf dataset?

**A10:** Thank you, we find this comment very helpful! It is important to distinguish unambiguously between rainfall used for adjusting and reference rainfall used as a "ground truth". We use the term "reference" for the rainfall to which we compare the best estimates from the adjusted CMLs. In the case of Prague, these are RGs located in the catchment, in the case of Dübendorf reference rainfall is taken as rainfall detected by the disdrometers along the CML path. In the first analysis where the effect of rainfall aggregation is investigated, we use the same RGs (resp. disdrometers) for adjusting and the same RGs as reference. We only used them at different temporal scales. In the second analysis (on Prague dataset only), where influence of RG spacing is tested, three different spatial layouts are used for adjusting, however we still use the reference RGs in the catchment for performance evaluation of the estimates from adjusted CMLs, i.e. we use same reference rainfall as in the first analysis.

To clarify this issue, we created new subsection (2.2 Rainfall data), where we moved an information about reference rainfall used for validating the results and about rainfall data used for adjusting CMLs. We also explicitly stated there, how reference rainfall is calculated. Finally, we corrected typo in the description of RG layouts (now second paragraph in the section 2.2 Rainfall data) where we we mistakenly referred to Fig. 1, left instead of middle and vice versa.

We have also restructured performance assessment section and clearly stated in its beginning (p.8, lines 21-22) that "We evaluate the adjusting method by directly comparing QPEs to reference rainfalls (Fig. 1, middle and right), both with a temporal resolution of one minute."

**Q11:** In Section 3.1 the authors discuss the reasons why parameter fitting for shorter intervals yield better results than for longer intervals. I don't really agree with this discussion. What effectively happens when the length of the aggregation is increased is that the CML data receive more weight in determining the temporal evolution of the rainfall signal (relative to the gauges). Because either the same gauges (Dübendorf) or a gauge dataset that is

well-correlated to the gauges that are used for the parameter fitting (Prague; see top-right panel of Fig. 4) are used for verification, it is expected that the results are best if the weight of the gauges is largest (i.e., for the shortest accumulation intervals). So I don't think that you can actually draw conclusions about which accumulation interval is best suited for this method based on these analyses.

**A11:** In section 3.1 we do not investigate optimal temporal aggregation intervals as stated already in the performance assessment section (see comment 9). We only study, how model performance worsens with increasing aggregation interval and we try to relate it to the autocorrelation structure of rainfall. We are very much aware of the fact that shorter aggregation intervals give more weights to the gauges and therefore, the best performance has to be achieved by short intervals when same gauges are used for adjusting and evaluation. The optimal aggregation interval is investigated subsequently in section 3.2 where aggregation is used to improve spatial representativeness of RGs far away from the catchment, resp. CMLs.

In our view, this is a misunderstanding, which partly arises from the wrong cross-references to Figure 1, which is now fixed (please see also our previous response).

In addition, we added into performance assessment section a short paragraph explaining the goal of the time averaging analysis, which is "explore whether the proposed adjusting method can be used to disaggregate cumulative rainfall data, such as hourly or daily values, to one-minute data."

**Q12:** On p.9, lines 20-22 the authors state that using daily rainfall accumulations to fit the model parameters would minimize the effect of diurnal fluctuations in baseline level. I think the converse is true: in order to minimize the effect of diurnal fluctuations, the model parameters should be fitted on a time scale that is significantly shorter than a day so that this variability is actually captured.

**A12:** We agree and we have removed this statement.

**Q13:** On p.13, Section 4.2 the authors discuss how the distribution of the $\gamma$ parameter changes with aggregation interval. This is then related to the fact that the proposed model includes the effect of wet antennas. However, this effect should be more related to the $k_w$ parameter of the model, and not so much to $\gamma$. Of course, the two model parameters can compensate, and this would result in wider distributions of $\gamma$, but this is a purely an effect of the fitting procedure.

**A13:** We have a different opinion on this particular issue and presume this is rather a misun-derstanding caused by the unfortunate naming the offset parameter "$k_w$" (now $\Delta$) of model (2) (see reply to comment 7) and its imprecise description in the original manuscript on p. 7, lines 7-9. This might create the false impression that only the offset parameter is responsible for wet antenna attenuation (WAA) correction. It is important to note that the simplified model (2) should be interpreted as "a combination of WAA model and simplified standard power-law model" (p. 7, lines 10-12). Although Overeem et al. 2011 suggest that, for their 15 minute CML data, WAA can be satisfactorily modelled as a constant, the other authors suggest more complex models. Given our theoretical understanding, these should generally depend on rainfall intensity (e.g. Kharadly et al. 2001, Leijnse et al. 2008). Indeed, we found that WAA react very dynamically on changes in rain-fall intensity. Spraying the radomes of some radios in Prague showed a substantial dynamic response. Immediately after spraying, attenuation increased by about 5 dB, decreased to 2.5 dB after 1 minute, and was almost not observable any more after 3-4 minutes (Fig. R1).

[Figure]

**Fig. R1.** Wet antenna attenuation of about 5 dB for a 38 GHz CML almost disappears within 3-4 minutes after spraying the antenna radome during dry weather.

If WAA depends on rainfall intensity, the linear approximation of any WAA model which reflects this dependence then should be affected by compensation of the offset parameter by the slope parameter. This also explains (p. 14, lines 1-3) the discrepancy between γ parameters of model (2) and α parameter of model (1) suggested by ITU (2005). Such discrepancy was already reported by Fenicia et. al (2012) "who estimated for their 23 GHz CML values of α substantially lower than values suggested by ITU (2005)" (p. 14, lines 3-4).

To clarify the nature of the simplified model (Eq. 2) and avoid misunderstanding, we labeled $k_w$ as Δ instead. And also changed the description of the parameters when first introducing model (2). Third, we better explained that the simplified model combined linear approximations of both the rainfall retrieval model (1) and wet antenna model in section 2.4. Finally, we have also added one sentence to the end of discussion section 4.2: "Interestingly, lower values of γ parameter compared to parameter α makes even shorter CMLs relatively sensitive to rainfall and thus capable to detect even light rainfalls"

**Q14:** On p.13, lines 17-18 the authors state that they've found a connection between the observed systematic errors and the degree of preservation of rainfall space-time structure through averaging. I don't really see this connection, and I think this should be better explained.

**A14:** We explained this connection in section 3.1 p. 9 lines 11-16 of the original manuscript, and we showed in the appendix A (and figure A1 in the manuscript) how increasing the aggregation interval smoothes out rainfall peaks and smoothes out the differences between low and high intense rainfall periods. In our opinion, this smoothing of rainfall peaks most likely explains why the identification of model (2) parameters worsens with increasing aggregation intervals. Although we did not formally describe the relation between preservation of correlation patterns in aggregated rainfall and model parameter identification, we sufficiently demonstrate that this relation exist and thus we can explicitly state on p. 14 lines 23-25 that our results suggest that the underestimation of peak intensities is influenced by the preservation of autocorrelation in the aggregated rainfall (Fig. A1, in the manuscript).

**Q15:** On p.14, line 9 the use of CML networks in sparsely gauged regions is mentioned. However, the method presented in this paper probably won't work in sparsely gauged reasons because rain gauges located close to the links are essential (see Figures 1, 4, and 5). So I think this statement needs to be altered.

**A15:** Thank you for the comment. It is also discussed on p. 15 lines 9-15, however, we agree that the presented analyses rather is a proof of concept than enables us to generalize and extrapolate to different conditions, e.g. RG layout, topology, climate, weather, etc. In particular extrapolation to sparsely gauged regions has to be performed with great care. We, therefore, modified the first sentence of the Conclusion section to: "Commercial microwave links (CMLs) can improve resolution of existing rain gauge and radar networks, especially in populated areas where there are often very dense."

**Q16:** On p.15, line 18 it is stated that CML networks can provide rainfall data on a (sub-)kilometer scale. However, I really don't think that this will be attainable with the method presented here. This is because of the fact that the CML data are adjusted to a (point) rain gauge somewhere in the vicinity, which will effectively smooth out much of the variability captured by the individual links. So this statement should also be put into perspective.

**A16:** Thank you for this comment, we have considered it carefully. Nevertheless, to our opinion combined use of RGs and CMLs can provide "insight into rainfall space-time structure at (sub)minute and (sub)kilometer scale" (p. 16 lines 24-26). We have demonstrated in presented analyses that even CMLs with sub-kilometer path lengths are, after adjusting, capable to provide accurate rainfall estimates outperforming RGs used for adjusting. In our investigation we poll CMLs with approx. 10 s resolution and it is technically possible to poll CMLs at (sub)second resolution (e.g. Chwala et al. 2016), although this might also be influenced by the firmware and hardware of the radio. Moreover, the CML networks especially in city centres can be very dense, in the Prague (CZ) city centre it is up to 50 CMLs per km$^2$. We, therefore, think it is appropriate to conclude that CMLs can provide "insight into rainfall space-time structure at (sub)kilometer and (sub)minute scale", although we are aware of the fact that adjusting can lead to averaging of rainfall peaks etc. This is, however, also happening when adjusting weather radar rainfall data and they are commonly used to estimates rainfall space-time structure at subkilometer scale.

**Q17:** In Figure 1, right panel, there seem to be white letters over the figure that are partly over the disdrometers.

**A17:** Thank you, we have corrected this.

**Q18**: In Figures 2, 5, and 6 the coefficient of determination ($R^2$) becomes negative. It would be good to give the definition of $R^2$ that was used in the paper in Section 2.6 (there are different versions of R$^2$, some of which cannot become negative).

**A18:** Thank you. We used pseudo $R^2$ as defined by Efron (1978), i.e. it is defined identically as the Nash-Sutcliffe efficiency (NSE), a popular measure in (urban) hydrology. We have changed the label in the whole manuscript including figures to NSE to avoid misunderstanding.

**Q19:** In Figures 3 and 4 the slope of the regression line y = ax (i.e., with fixed offset) is given. It should be noted here that the correlation between the two variables affects this slope. The slope will always be lower with a low correlation coefficient (you can try this by switching the x- and y-axes; see also the right-hand panels of Fig.4).

**A19:** This is another valuable remark. We have added correlation coefficients into the legends of scatterplots in both figures (Fig. 3 and 4). The correlation coefficients on Figure 4 shows, that even CMLs adjusted to remote RGs correlate very well with reference rainfall. The slope of CML reference rainfall regression line is therefore rather influenced by systematic underestimation of rainfall peaks. In contrast to that, the correlation between RG layouts which cover larger areas and reference rainfalls are much lower (at 1 min resolution), which indeed affects the slope of regression lines. However, aggregating the RG intensities over longer intervals increases the correlation. Consequently, a longer aggregation interval improves the performance of the adjusting algorithm compared to shorter aggregation (see Fig. 5 in the manuscript: in the case of layout B2 with relatively distant RGs - the NSE of 1h ranges between 0.50-0.91 with median 0.78, whereas 5 min only achieves NSE between 0.2-0.86 with median 0.75). The areal averaging leads, however, to the smoothing out of rainfall peaks which in turn systematically affects peak rainfalls estimated from adjusted CMLs.

When we were preparing revised figure 3, we found that we presented for Prague dataset performance of CML no. 1 instead of mean of all four CMLs as stated in the figure caption. This is now corrected and figure 3 presents mean CML rainfall from all four CMLs.

**Reply to Reviewer 2: S. Thorndahl**

**General comments**

Reviewer: The manuscript provides methods for adjusting rainfall estimates from commercial microwave links (CMLs) to rain gauges (RGs). It compares different temporal scales for adjustment and different layouts of gauge/CML systems. The work is novel and addresses very relevant issues in high resolution rainfall estimation in urban areas. It is well written and understandable and would fit well into the scope of HESS. Although not an expert in CMLs (but in radar rainfall estimation), I have some comments and suggestions which in my opinion could improve the manuscript.

Authors: First, we would like to thank reviewer for all the remarks and recommendations how to complete the manuscript and improve its clarity. Clearly, an expert on weather radars experienced in adjusting to rain gauges can give substantial advice.

**Specific comments**

**Q1:** It is unclear whether the paper aims for on-line (real-time) adjustment of CML's and thus real-time rainfall estimation or to estimate historical rainfall. Real-time adjustment would be associated with larger uncertainties.

**A1:** In our analysis we assess the method in the setting for historical rainfalls. However, the method does not "look into future" when continuously adjusting model (2), but uses rainfall intensity from the time point for which the adjustment is done and then from several time points in the past (p. 8, lines 9-11). Thus, method can be used with additional tuning in near real-time setting.

To have a better evidence base, we performed additional analyses where CML rainfall retrieval model is continuously adjusted based only on past data. The results of these analyses confirmed that use of the method in real-time setting leads to only slightly worse CML performance in comparison to the original analysis. We therefore explicitly stated in the section 2.5 that we assess the method in the setting for historical rainfalls and also added into section 2.4 this information: "When conditioning model (2) on historical data, parameter ranges can be set on the basis of the whole available dataset. For real-time application ranges has to be estimated from past periods." Furthermore, we added into Discussion section subsection 4.3 discussing use of the adjusting algorithm in real-time setting.

**Q2:** P4L31-P5L3: This is almost a conclusion of the paper. It does not belong in an introduction – but could be applied in the abstract.

**A2:** Thank you for this comment. In our view, this paragraph improves the intelligibility and clarity of a manuscript to i) have a very specific message and ii) convey the message to the reader. This can include explicitly stating the novelty of the work, but also concrete results. Then, a reader is not left in the dark what to expect and will not have major surprises - which are always confusing - during reading. As the abstract is too short, this info can go into the introduction. In our view, the redundancy of information-pieces (twice mentioned in the abstract and the intro) is a small price to pay for the increased clarity.

**Q3:** In section 2, it should be argued why two different experimental sites are used. Could the same results not have been derived using only one site – or is there an objective to compare the two sites in terms of data, layout, etc.

**A3:** Thank you for this comment. The reasons behind using data from two sites were distributed over the two sections 2.1.1 and 2.1.2 (differences in operational mode P5L16-17, P5L22-24, P6L1-2, or P6L12-13) and they included different reference rainfall data and power-control settings of CMLs on each of the sites. We agree that this is sub-optimal.

We have therefore briefly mention our main reason in the introductory paragraph of the section 2.1: "We analyse datasets from two different experimental sites, Dübendorf (CH) and Prague-Letnany (CZ). The dataset from Dübendorf contains detailed reference rainfall measurements along a CML path, which provide an excellent basis for investigating a rainfall from a single CML. In contrast, the areal rainfall observations from Prague are more appropriate to analyse rainfall retrieval from multiple CMLs and thus more relevant to evaluate the proposed adjusting method for common urban hydrological applications."

**Q4:** During the paper it is also a bit confusing where averages of CMLs are used (in Prague) and when single CMLs are used. Please be clearer on this.

**A4:** Thanks, we now see that this information was indeed missing in the method section (2.6 performance assessment) but was presented it in the Result section (P93-4 and P9L31-P10L3 in the old manuscript) instead. We have therefore provided this information in the new subsection 2.2 Rainfall data.

**Q5:** P6 bottom. It is unclear how you define an event. This is not necessarily an easy task operating with more than one rain sensor. Please clarify.

**A5:** The events at both experimental sites are first defined from each of the sensors and then the event periods are merged by simply increasing the duration to include the very first and the very last observation of a sensor. In the case of Dübendorf the events were defined based on disdrometer classification. In the case of Prague, events were defined from reference RG measurements. The beginning of an event is assumed to be 15 minutes before the first tip of RG and end of event 15 minutes after the last tip. In addition, the beginnings and ends of the events in Prague area were rounded (down resp. up) to full hours to ease the analysis with aggregated rainfall intensities. At both sites two rainy periods separated by shorter interval than 30 minutes are assumed to be the same event. Note, however, that adjusting is performed over whole experimental period and thus it is independent of event definition. The event definition therefore influences the performance evaluation, i.e. by the (non-)selection of events.

We have added an information about event definition into section 2.2 Rainfall data and also stressed at the beginning of this section 2.5 Performance assessment that "QPEs are adjusted over the whole experimental periods but evaluated only for rainfall events, which exceeded 5 mm in total, i.e., they are relevant for stormwater management (Table 1)."

**Q6:** Section 2.6. You state that you adjust on different aggregation levels ranging from 5 min to 1 day, but compare on 1 minute values. Couldn't there be reason also to compare on larger aggregation levels than 1 min. It is well known that for small rain intensities rain gauges are not very accurate. E.g. one tip of 0.1 mm per minute

in a tipping bucket rain gauge corresponds to 6 mm/h. An error of +/- 6 mm/h on gauge estimates over one minute for intensities larger than 6 mm/h, it therefore not unrealistic. For smaller intensities where the intensities are estimated using the time between two tips, the intensity at minute scale might be somewhat uncertain. In a paper (Thorndahl et al. 2014) we made radar-rain gauge adjustment over different temporal scales, but also compared the results over different scales. Maybe you could find some inspiration here.

**A6:** Thank you for this suggestion. We used one minute reference data because this is the temporal resolution required for rainfall-runoff modeling at the scale of small urban catchments and our long-term intention is to provide rainfall data which could be used for this purpose. In the case of Dubendorf site, disdrometers are very well suited for providing rainfall data at 1 min resolution. The sampling error of tipping bucket RGs in Prague is partly reduced by calculating areal rainfall from six RGs relatively close to each other. Nevertheless, we agree that this sampling error may influence *NSE* values. We have therefore compared rainfall estimates from CML data also at other temporal scales (Fig R2, this response).

[Figure]

**Fig. R2.** Comparison of NSE for different temporal aggregations of reference rainfall. Mean of four CMLs (see figure 2 of the original manuscript) adjusted to rainfall having different aggregation intervals.

We find only small changes in *NSE* values when comparing CML rainfall to reference rainfall at larger temporal scales. This indicates that the analysis even at 1 min scale is not substantially influenced by random errors. Interestingly, we see a slight increase in *NSE* values for larger temporal scales of reference rainfall, although aggregation should reduce RG sampling errors. In our view, giving a larger weight to the RG data in the adjusting procedure increases the *NSE*, because the temporal scale of reference rainfall gets closer to the aggregation interval used for CML adjusting (for details see comment no. 11 of the reviewer 1). The results presented here do not, however, change our conclusions drawn in the original manuscript where we only presented the performance for 1 minute data. We therefore think it is sufficient to show only these results.

**Q7:** With regards to estimating area rainfall (section 2.2 and 3.2) I guess results are still compared on the minute scale and adjustment is performed on larges temporal scales. I guess this will be associated with many random errors if there is rain in one gauge and not in another? Again I suggest to also comparing e.g. hourly estimates of Rainfall.

**A7:** Yes, the discrepancy between rainfall measured by those RG layouts which were used for adjusting and those used for validation purposes (reference rainfall), indeed influences the performance of the adjusting procedure. Here, we reduce these errors by aggregating the RG data used for adjusting to longer intervals (up to 1 h). The

performance of the procedure is then evaluated by comparing adjusted CMLs to the reference rainfall (i.e. RGs in the catchment). Thus, all rainfall observation errors which stem from RG layouts (including instrumental errors, sampling errors and limited spatial representativeness of point RG measurements) are implicitly included in the evaluation. The comparison at larger time scales would indeed reduce the sampling error in reference rainfall. However, as discussed in the reply above, their influence on the performance is small. Moreover, our adjusting procedure is only relevant where the temporal scale of reference rainfall (resp. adjusted rainfall) is finer than the aggregation interval used for CML adjusting. As we identified in our analysis that optimal aggregation intervals for the evaluated RG layouts are relatively short (15 min for layouts A1 resp. B1 and 1 h for layout B2), the comparison to e.g. hourly estimates is not useful.

**Q8:** Related to my comment no 4. I think it would be interesting to see a scatter plot of a single CML vs a single RG and how $R^2$ would depend on the range between CML and RG?

**A8:** Thank you for this suggestion. Unfortunately, although this might be an interesting analysis our experimental layout is not suitable for that. Each CML included in the analysis (see Fig. 1 in the manuscript) has different features (lengths, frequencies, polarizations) which considerably influences its performance in terms of rainfall estimation. The differences between single CMLs and a corresponding RG would be dominated by these differences. In our experience, the discrepancy between path integrated and point measured rainfall usually dominates the discrepancy due to different locations of the CML and the RG.

**Q9:** For the Dübendorf site it is unclear what you use the disdrometers for. Don't you use the RGs for adjustment/validation? Related to the problem above, disdrometers might be more accurate for small rain intensities?!

**A9:** Thank you, we have modified the sentence at P5L27-28 to: "In addition, three tipping bucket RGs measure rainfall intensities which make it possible to validate the disdrometer data." Furthermore, we explain in the section 2.2 Rainfall data (P6L25-26) that "To validate the QPEs from CMLs, we use different reference rainfall information. For Dübendorf we take the mean of five disdrometers along the CML path (Fig. 1, right) and for Prague-Letnany the mean of the six RGs (Fig. 1, middle)."

**Q10:** P9L18-19. A likely reason for the smaller scatter on the 1 day aggregation levels might be found in the fact that all of your events (except one) have duration shorter than 1 day. Thus, for some events same results for 12 and 24 h should be expected!

**A10:** In our study we adjusted each CML over the whole experimental period, although it is evaluated only on events listed in the table 1. Thus for longer aggregation intervals also other events (with heights lower than 5 mm) influence the adjusting. This is also one of the reasons why CML adjusted with 12 h aggregation interval have different scatter than adjusted to 1 d aggregation interval.

We have therefore explained more clearly in the revised section 2.5 (Performance assessment) that: "CML QPEs are adjusted over whole experimental periods but evaluated only for rainfall events, which exceeded 5 mm in total and thus are relevant for stormwater management (Table 1)."

**Q11:** Figure 1. Please use lat/long or UTM rather than a local coordinate system.

**A11:** Thank you for the suggestion. We changed the coordinate system to UTM in the figure 1.

**References:**

Thorndahl, S., Nielsen, J.E., Rasmussen, M.R., 2014. Bias adjustment and advection interpolation of long-term high resolution radar rainfall series. Journal of Hydrology 508, 214–226. doi:10.1016/j.jhydrol.2013.10.056

[revised manuscript text omitted]